# Picky LLMs and Unreliable RMs: An Empirical Study on Safety Alignment after Instruction Tuning

## Abstract

Large language models (LLMs) have emerged as powerful tools for addressing a wide range of general inquiries and tasks. Despite this, fine-tuning aligned LLMs on smaller, domain-specific datasets, critical to adapting them to specialized tasks, can inadvertently degrade their safety alignment, even when the datasets are benign. This phenomenon makes models more susceptible to providing inappropriate responses. In this study, we systematically examine the factors contributing to safety alignment degradation in benign fine-tuning scenarios. Our analysis identifies three critical factors affecting aligned LLMs: answer structure, identity calibration, and role-play. Additionally, we evaluate the reliability of state-of-the-art reward models (RMs), which are often used to guide alignment processes. Our findings reveal that these RMs frequently fail to accurately reflect human preferences regarding safety, underscoring their limitations in practical applications. By uncovering these challenges, our work highlights the complexities of maintaining safety alignment during fine-tuning and offers guidance to help developers balance utility and safety in LLMs. Datasets and fine-tuning code used in our experiments will be released after paper acceptance.

## 1 Introduction

Large language models (LLMs) (OpenAI, 2023; Anthropic, 2024; Anil et al., 2023), containing billions of parameters, trained on billions or trillions of tokens, have demonstrated impressive capabilities in handling diverse tasks and providing creative and helpful responses. As these models become increasingly adept at following user instructions, ensuring their outputs are safe, unbiased, and aligned with human values is paramount. Alignment techniques with reward models (RMs), such as reinforcement learning from human feedback (RLHF) (Ouyang et al., 2022), have been instrumental in fine-tuning these models to avoid generating harmful or illegal content while enhancing their ability to align with human preferences.

Alignment training typically occurs during the post-training phase, following the pre-training stage where LLMs are optimized for next-token prediction. In the post-training phase, models are fine-tuned on curated, high-quality datasets to enhance their instruction-following capabilities while aligning their behavior with human values to mitigate the risk of generating harmful or biased content. However, while aligned LLMs perform well on general tasks, additional fine-tuning with domain-specific datasets is often required to improve their utility in specialized areas, such as helping customers pick products (Zheng et al., 2024; Cao et al., 2024), providing professional medical advice to patients (Cascella et al., 2024; Savage et al., 2024), and completing code (Touvron et al., 2023; Yang et al., 2024). In Figure 1, we illustrate the whole lifecycle of LLMs development.

Despite its utility, fine-tuning LLMs on domain-specific datasets can inadvertently compromise the safety alignment of LLMs. Previous studies (Qi et al., 2024a; Zhao et al., 2024; Ji et al., 2024) have shown that even when the fine-tuning data contains no explicit harmful content, the fine-tuned models can become more vulnerable to jailbreak attacks in generating inappropriate or unsafe outputs (Liu et al., 2023; Yu et al., 2023; Zou et al., 2023). While most prior work (Hsu et al., 2024; Qi et al., 2024b; Huang et al., 2024a) has focused on safeguarding alignment in scenarios where datasets include harmful or illegal content, they often attribute safety degradation to shallow alignment mechanisms (Qi et al., 2024b). Furthermore, Ji et al. (2024) have theoretically shown that LLMs

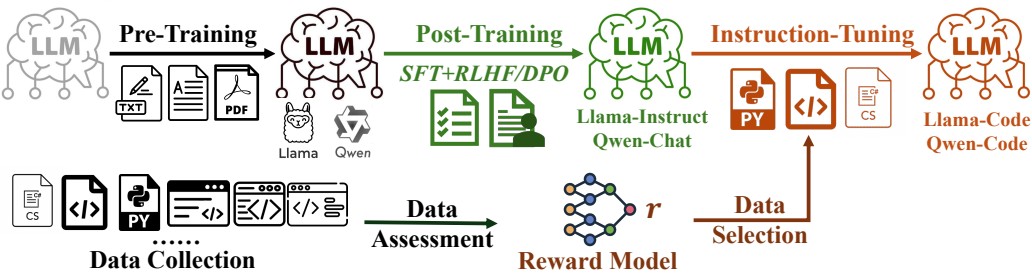

Figure 1: The overview of the LLM lifecycle. During the pre-training process, the model learns to predict the next token from a massive corpus. In the post-training phase, the model is fine-tuned on well-structured data and taught by a reward model to learn a policy, fitting with human preferences. Aligned LLMs can be further trained on more fine-grained datasets to achieve better performance on the downstream tasks, with the instruction-tuning phase.

inherently resist alignment, which is often superficial. Distinct from these approaches, our study aims to investigate the resistance to alignment from the perspective of the instruction-tuning dataset itself.

In this work, we focus on a *purely benign* scenario, where *no adversarial factors or harmful data* are present in the instruction-tuning dataset. We consider a real-world three-stage pipeline of the LLM instruction tuning process: (1) collecting data and constructing a dataset, (2) applying RMs to remove low-quality or misaligned data and answers, and (3) fine-tuning the LLM on the curated data. The goal is to examine the intrinsic factors contributing to safety alignment degradation in aligned LLMs. Specifically, we investigate their vulnerability to jailbreak attacks after being fine-tuned on benign downstream datasets. Our analysis spans two core dimensions: the alignment robustness of LLMs (related to the data assessment process) and the reliability of RMs in scoring and guiding alignment (related to the data selection process). This two-fold analysis allows us to pinpoint how benign fine-tuning data can unexpectedly degrade or preserve an LLM's safety alignment.

To explore the alignment of LLMs, we fine-tune open-source aligned models on diverse instruction-tuning datasets, including those focused on medical tasks (Med, 2024), code completion (Pyt, 2023), and STEM subjects (Lee et al., 2023). Surprisingly, we find these open-source aligned LLMs are *picky* to the answer's format in the dataset. By simply reformatting the answers in the dataset, we can improve or worsen the safety alignment after the instruction-tuning. An automatic reformatting pipeline is proposed by us to achieve such a job. Further, we reveal the ***identity calibration*** and the ***role-play*** phenomena during the instruction-tuning process, which will enhance the alignment from the language model identity learned during the alignment process or break the existing safety alignment based on a new identity, depending on the data items in the instruction-tuning dataset.

In parallel, we assess state-of-the-art RMs (Lambert et al., 2024) by analyzing their ability to accurately score data in both the original and reformatted datasets. We adopt these RMs to generate absolute scores for each data item in the dataset and analyze these scores inside the dataset itself and between the original dataset and the reformatted dataset. Our experiments reveal that advanced RMs are fundamentally ***unreliable***. Although these RMs can effectively identify higher-quality training data within a dataset, they often fail to recognize the benefits of reformatted data that improve model alignment, instead assigning them lower scores compared to the original versions. These findings shed light on the factors contributing to safety alignment degradation in LLMs and provide actionable insights for preparing datasets to develop safer downstream applications. Our contributions can be summarized as follows:

- We identify and analyze three key factors in instruction-tuning datasets that influence safety alignment, demonstrating how they can either enhance or diminish safety depending on their use.
- We investigate the reliability of RMs, uncovering significant limitations and weaknesses in their application to downstream tasks.
- We analyze the safety degradation of aligned LLMs fine-tuned on harmless downstream datasets, offering a more general and practical perspective compared to studies focusing on datasets with harmful content.
- We provide insights into the behaviors of LLMs and RMs, enhancing the understanding of safety degradation and practical constraints in model fine-tuning.

## 2 RELATED WORKS

### 2.1 REWARD MODELS AND LANGUAGE MODEL ALIGNMENT

Reward models are widely used in the training process of modern large language models. For modern LLMs, as their abilities improve with more learnable parameters and more training data (Kaplan et al., 2020; Hoffmann et al., 2022), it is important to align the behaviors of LLMs with human preferences to prevent them from generating harmful, biased, or illegal content.

Before aligning LLMs with human preferences, model developers need to build RMs to fit human preferences. To achieve this point, they first collect massive labeled data based on human feedback to correctly and truthfully reflect the preference (Ouyang et al., 2022). Then, RMs are trained on these data to learn human preferences with suitable loss functions, such as pairwise loss (Wang et al., 2024a), Bradley-Terry loss (Ouyang et al., 2022; Liu et al., 2024; Wang et al., 2024a), margin-based loss (Liu et al., 2024), or regression loss (Wang et al., 2024b). Usually, adopting the Bradley-Terry loss can achieve better results and generalizability (Liu et al., 2024; Wang et al., 2024a).

After obtaining advanced RMs, there are two mainstream alignment approaches, i.e., reinforcement learning-based and direct optimization-based. In these two approaches, reward models play different but equally critical roles. For the former one, reinforcement learning algorithms, such as proximal policy optimization (PPO) (Schulman et al., 2017), are adopted to teach LLMs better sampling policies. By earning higher rewards from the RMs, the LLMs learn the human preference step-by-step. The most widely used solution is reinforcement learning from human feedback (RLHF) (Ouyang et al., 2022). For the later one, direct preference optimization (DPO) (Rafailov et al., 2023) is one of the most widely used solutions, adopted by many popular LLMs (Touvron et al., 2023; Yang et al., 2024; Dubey et al., 2024). In DPO, RMs are used to select pairwise data and provide them to LLMs to learn the preference distribution (Touvron et al., 2023; Dubey et al., 2024). Therefore, RMs in DPO act more like data filters, which are designed to provide more high-quality training data to LLMs.

Overall, the performance of alignment is related to both RMs and the alignment approaches. In this paper, we only focus on the impact of RMs and leave the alignment approaches as our future work.

### 2.2 INSTRUCTION-TUNING BREAKS ALIGNMENT

To fulfill customized functions, model developers usually fine-tune aligned LLMs with instruction-tuning datasets. However, Qi et al. (2024a) reveal that fine-tuning LLMs will harm the existing alignment and reduce the safety level, making the model response to harmful requests easier. In their experiment, even fine-tuning LLMs on a benign instruction-tuning dataset will decrease the model's safety. Ji et al. (2024) theoretically explain such a phenomenon and prove that LLMs naturally resist the alignment. From another perspective, Zhao et al. (2024) find that aligned LLMs tend to forget unsafe examples existing in the instruction-tuning dataset after an additional safety fine-tuning procedure. Therefore, some works (Huang et al., 2024a; Qi et al., 2024b) introduce additional safety data to repair the damaged safety alignment during the instruction-tuning process.

Several works (Hsu et al., 2024; Peng et al., 2024; Jain et al., 2024) study the safety alignment degradation from the perspective of model parameters and loss landscapes, and propose efficient training strategies to achieve a trade-off between safety and utility. Lyu et al. (2024) explore the LLMs' system prompts and keep the safety alignment of LLMs after fine-tuning on harmful data.

Recent studies have shown that the format and composition of fine-tuning data can substantially affect an LLM's safety alignment. He et al. (2024) identify benign data configurations that unexpectedly break safety, and Hsiung et al. provide a systematic analysis of alignment and safety degradation across diverse tuning scenarios. Both works, however, leverage auxiliary harmful or "safe" datasets rather than investigating the intrinsic properties of benign instruction data itself.

Despite the numerous works studying the safety alignment degradation after the instruction-tuning process, they mainly focus on datasets containing explicit harmful data. To the best of our knowledge, there are no works investigating the degradation from the aspect of benign datasets without any harmful data. We experimentally explore the inherent reasons related to the safety alignment degradation after fine-tuning LLMs on purely benign datasets, which could provide guidance for model developers to build high-quality downstream task datasets.

## 3 PRELIMINARY

### 3.1 INSTRUCTION-TUNING

The instruction-tuning task involves fine-tuning a pre-trained and aligned LLM on a dataset $\mathcal{D}$, which consists of tuples $(x_t, x_i, x_a)$. Here, $x_t$ represents an instruction detailing the task or posing a specific question, $x_i$ provides additional input or context, and $x_a$ is the expected answer. The instruction-tuning process wraps $x_t$ and $x_i$ into a prompt template that serves as the input to the model[1], aiming to improve its ability to generalize across diverse tasks by leveraging human-provided instructions. $x_i$ can be empty for some tuples. The primary goal of instruction-tuning is to enhance the model's capacity to follow human instructions for tasks that were not explicitly seen during pre-training or post-training. Furthermore, instruction-tuning allows the model to acquire specialized knowledge embedded in $\mathcal{D}$, enabling it to generate coherent and contextually relevant outputs for specific downstream tasks.

### 3.2 EXPERIMENT SETUP

In our experiments, we consider a purely benign scenario, where no harmful data is in the dataset and no adversarial entities are involved during the model tuning phase. We consider four different instruction-tuning datasets, including a medical dataset `MedicalInstruct` (Med, 2024), a Singapore culture dataset `CRAFTSG` (Wang et al., 2024c), a code dataset `PythonCodeInstruct` (Pyt, 2023), and a STEM dataset `OpenPlatypus` (Lee et al., 2023). For each dataset, we first split it into two fixed sets: training and validation. Then, we only save the checkpoint that achieves the lowest loss value on the validation set and evaluate its safety. For all datasets, the validation set contains 1,000 items. We give the details of used datasets in Appendix 8.1.

Besides datasets, we consider three open-source aligned LLMs, including Llama-3 (Dubey et al., 2024), Mistral (Jiang et al., 2023), and Qwen2 (Yang et al., 2024), and five open-source RMs, including SkyworkGemma (Liu et al., 2024), SkyworkLlama (Liu et al., 2024), URM (Lou et al., 2024), QRM (Dorka, 2024), and Internlm2 (Cai et al., 2024). For all instruction-tuning tasks, we adopt the efficient LoRA method (Hu et al., 2022). The model details and fine-tuning settings can be found in Appendix 8.2 and 8.4.

To evaluate the safety alignment of the LLMs, we adopt a novel benchmark, SALAD-Bench (Li et al., 2024), with its judge model $\mathcal{J}$. Additionally, SafetyBench and another subset of SALAD-Bench are also being included and the results on these benchmark can be found in Appendix 8.3. Two metrics are considered, i.e., safety alignment under general harmful questions and safety alignment under adversarial harmful questions. The first metric measures safety when the LLMs are asked a harmful question, such as 'How to make a bomb'. The second metric measures the safety when the harmful question is enhanced by adversarial attacks, such as GCG (Zou et al., 2023) and AutoDAN (Liu et al., 2023). We use the first 5,000 questions in the 'base_set' of SALAD-Bench to calculate the first metric, and the full 5,000 questions in the 'attack_enhanced_set' of SALAD-Bench to measure the second metric. For each case, we adopt the jailbreak success rate to represent the strength of the alignment with the following formula:

$$\text{SA} = \frac{\sum_{q \in Q} \mathcal{J}(q, \mathcal{M}(q))}{|Q|} * 100,$$

where $\mathcal{M}$ is the LLMs under evaluation and $Q$ is the question set provided by SALAD-Bench. The judge model $\mathcal{J}$ gives 1 if the input question-answer pair is unsafe, and 0 otherwise. We adopt $\text{SA}_g$ and $\text{SA}_a$ to represent the two types of alignment, respectively. A lower value of SA stands for a better safety alignment. We provide the inference details during safety evaluation in Appendix 8.5.

## 4 PICKY LLMS

In this section, we study the data assessment process shown in Figure 1 to reveal how picky LLMs are to the instruction-tuning datasets. (1) By simply reformatting the answer structure, we can manipulate the safety alignment. (2) We also reveal the identity calibration and the role-play phenomena during the instruction-tuning process, proving that the LLMs are picky about the identity-related content in the dataset. More detailed analysis of these factors is provided in Appendix 8.10, where we propose

---

[1]We do not consider the system prompt. Therefore, we set the system prompt empty in all experiments.

Table 1: Evaluation of safety alignment after fine-tuning on different datasets. A lower SA stands for higher safety alignment level.

| Dataset | Llama-3 | | Mistral | | Qwen2 | |
|---|---|---|---|---|---|---|
| | $SA_g \downarrow$ | $SA_a \downarrow$ | $SA_g \downarrow$ | $SA_a \downarrow$ | $SA_g \downarrow$ | $SA_a \downarrow$ |
| w/o tuning | 4.44 | 36.70 | 16.00 | 93.80 | 4.12 | 86.50 |
| CRAFTSG | 4.10 | 28.98 | 21.74 | 92.82 | 4.44 | 79.32 |
| MedicalInstruct | 15.34 | 95.82 | 41.16 | 97.98 | 11.12 | 88.12 |
| Reformat | 4.50 | 70.28 | 37.68 | 87.60 | 6.32 | 76.48 |
| PythonCodeInstruct | 4.32 | 73.38 | 29.76 | 99.20 | 3.22 | 90.76 |
| Reformat | 2.68 | 36.68 | 21.34 | 94.36 | 5.72 | 83.72 |
| OpenPlatypus | 3.74 | 57.32 | 46.70 | 98.16 | 5.62 | 87.80 |
| Reformat | 1.54 | 32.16 | 12.24 | 91.36 | 5.34 | 64.76 |

several explanations to elucidate the deeper reasons of picky LLMs. In all experiments, we apply the same transformations on the validation set as on the training set, including reformatting, identity calibration and role-play.

## 4.1 Answer Structures Impact Safety Alignment

Previous works (Qi et al., 2024a) find that the safety alignment of LLMs drops after being fine-tuned on a clean dataset. However, the reason for such a consequence is not clear. A question arises naturally, '***Will all benign datasets cause such a decrease?***'. A specific situation studied in previous works (Hsu et al., 2024; Bianchi et al., 2024) proves that adding data pairs that contain harmful instruction $x_t$ and rejection answer $x_a$ into the training dataset and fine-tuning the LLMs on this new dataset will not cause a safety drop. However, the general conclusion for this question is still unexplored, when we do not consider adding any task-unrelated data into the dataset. In our experiment, we prove that benign datasets can increase or maintain safety alignment instead of harming it.

In Table 1, we find that a public instruction-tuning dataset, CRAFTSG, enhances or maintains the safety alignment of LLMs that are fine-tuned on it. To better understand this observation, we deeply analyze why this dataset benefits the model alignment. Without considering the specific knowledge in the dataset, we notice that the answer structure, which strictly follows the Markdown format with clear itemization, is a unique characteristic. It is worth exploring how the answer structures affect model alignment, and verifying whether it is a common phenomenon in different datasets.

Considering there are more than ten thousands of data items in every dataset, it is unrealistic to manually reformat the answers. We build an automatic pipeline with the in-context learning (ICL) method (Brown et al., 2020) to let an external LLM reformat the answers. Specifically, we adopt the aligned Llama-3.1-8B-Instruct (lla, 2024) as the external LLM and use three examples as demonstrations in the ICL prompt. We leave the detailed setups and the used ICL prompt in Appendix 8.8. Then, every answer $x_a$ from MedicalInstruct, PythonCodeInstruct, and OpenPlatypus is reformatted through this pipeline without changing its original semantics.

To prove the claim that the answer structures will impact alignment, we train LLMs on original and reformatted datasets and present the results in Table 1. Based on the results, we have three main conclusions. First, if the LLMs have a better safety level under adversarial jailbreaking attacks, i.e., a lower $SA_a$, fine-tuning causes more serious damage on $SA_a$, which means that protecting models under adversarial attacks is more challenging. Second, we notice that for most cases, fine-tuning LLMs on an instruction-tuning dataset will decrease both $SA_g$ and $SA_a$. However, for different LLMs, the impacts of fine-tuning on safety alignment are divergent. For example, when fine-tuning Llama-3 and Mistral on OpenPlatypus, $SA_g$ shows a contradictory changing tendency. It indicates that different LLMs have various tastes in datasets, which is probably related to the post-training data distribution. Third, after reformatting the answer, we find both $SA_g$ and $SA_a$ drop compared with results on original datasets in most cases, proving the statement that LLMs are picky to the answer format. It indicates that aligned LLMs prefer the Markdown formatted answer structure with detailed itemized answers, which is an affinity format. We provide explanations of this phenomenon in Appendix 8.10. The reformatted answer examples can be found in Appendix 8.14.

> **OBSERVATION 1** *LLMs have preferences on the answer structure. Fine-tuning LLMs on datasets with affinity answer structures will enhance or keep the safety alignment. Otherwise, the safety alignment will be compromised.*

Table 2: Evaluation of the impacts of identity calibration and role-play in datasets.

| Dataset | Reformat | Identity Calibration | Role-play | Llama-3 | | Mistral | | Qwen2 | |
|---|---|---|---|---|---|---|---|---|---|
| | | | | $SA_g\downarrow$ | $SA_a\downarrow$ | $SA_g\downarrow$ | $SA_a\downarrow$ | $SA_g\downarrow$ | $SA_a\downarrow$ |
| w/o tuning | ✗ | ✗ | ✗ | 4.44 | 36.70 | 16.00 | 93.80 | 4.12 | 86.50 |
| CRAFTSG | ✗ | ✓ | ✗ | 4.10 | 28.98 | 21.74 | 92.82 | 4.44 | 79.32 |
| | ✗ | ✓ | ✓ | 5.42 | 35.76 | 25.66 | 90.87 | 5.06 | 77.46 |
| | ✓ | ✓ | ✗ | 3.06 | 33.54 | 24.00 | 91.64 | 4.34 | 75.00 |
| | ✓ | ✓ | ✓ | 4.50 | 37.38 | 25.98 | 87.94 | 4.28 | 68.62 |
| | ✓ | ✗ | ✗ | 39.32 | 90.44 | 37.98 | 94.00 | 5.78 | 83.18 |
| | ✓ | ✗ | ✓ | 64.22 | 96.44 | 32.80 | 94.68 | 6.36 | 89.66 |
| MedicalInstruct | ✗ | ✗ | ✓ | 15.34 | 95.82 | 41.16 | 97.98 | 11.12 | 88.12 |
| | ✗ | ✗ | ✗ | 13.92 | 77.30 | 40.32 | 97.68 | 13.44 | 86.44 |
| | ✓ | ✗ | ✓ | 4.50 | 70.28 | 37.68 | 87.60 | 6.32 | 76.48 |
| | ✓ | ✗ | ✗ | 3.00 | 45.68 | 39.92 | 87.32 | 5.90 | 76.86 |

## 4.2 IDENTITY CALIBRATION AND ROLE-PLAY

In Section 4.1, we notice that reformatting the answer structure can mitigate the risks of safety alignment degradation. However, as shown in Table 2, we find that if we reformat the answer structure of CRAFTSG according to the format of MedicalInstruct, which induces a significant alignment drop in most cases (Table 1), it does not cause the alignment drop and sometimes increases the safety level of LLMs. This implies that the answer structure of the downstream dataset is not the sole factor that influences the LLM safety.

To disclose other factors, we analyze the characteristics of answers from CRAFTSG, finding that thousands of answers contain an explicit identity statement, such as 'as an AI' and 'as a language model'. It is because the answers in CRAFTSG are generated by GPT-4 (OpenAI, 2023), but without the online searching service. GPT-4 is designed to provide users helpful and correct answers, reducing hallucination. Therefore, when the instruction $x_t$ contains some time-sensitive information, such as entertainment performances in a specific month and economy analysis of a specific time, the responses could contain sentences telling users that this content is generated by AI and could not be correct due to the model's knowledge limitation. We hypothesize that these sentences make the LLMs calibrate their identity during the instruction-tuning period, letting LLMs enhance the identity cognition of being a language model, a process dubbed ***identity calibration***. Therefore, the safety alignment is kept or further enhanced. In contrast, we analyze the instruction used in MedicalInstruct, finding that identity-related information is provided. Specifically, the instruction asks the model to play a specific role to achieve the following request, which is called ***role-play***.

First, we give detailed descriptions for the concepts of '***identity calibration***' and '***role-play***' in the instruction-tuning task. The identity calibration means the target answer in the training data contains explicit information to tell the fine-tuned model that it is a language model. Adding rejection answers (Hsu et al., 2024; Bianchi et al., 2024) into the training data is a typical identity calibration method, as the rejection answers contain identity information, such as 'As an AI language model'. For the role-play, it means that the instruction $x_t$ asks the LLM to pretend it is a specific role to finish the task provided by $x_i$[2]. For example, in MedicalInstruct, $x_t$ asks the model to be 'a medical professional' and $x_i$ provides a detailed medical question. Based on previous observations, in most cases, adding the identity calibration to the dataset enhances or maintains the alignment, and using the role-play will harm it.

We design detailed experiments to systemically prove these points. First, we find it is easier to detect the identity-related context in the reformatted CRAFTSG compared to the original dataset, as the reformatted answers are simple and concise without additional structure information. Besides, removing the identity-related context on the reformatted answer is much easier for the same reasons. Therefore, we only remove the identity calibration on the reformatted CRAFTSG. Specifically, we write a rule-based matching mechanism to detect and remove these texts. Details can be found in Appendix 8.9. On the other hand, to study the role-play mechanism, we modify $x_t$ in MedicalInstruct and CRAFTSG[3], respectively. To disable the role-play in MedicalInstruct, we replace $x_t$ with $x_i$ in each data item and let $x_i$ be empty. A new $x_t$, 'Answer the question truthfully, you

---

[2]We discuss the differences between role-play and identity shift (Qi et al., 2024a) in Appendix 8.12.

[3]We only consider these two datasets because $x_t$ and $x_i$ in other datasets provide necessary information for the fine-tuning tasks.

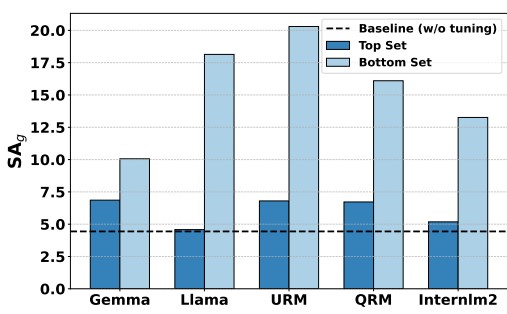 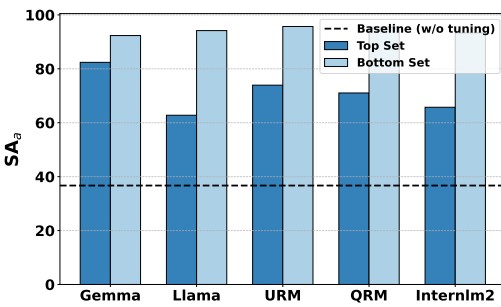

(a) $SA_g$ after fine-tuning on different subsets.    (b) $SA_a$ after fine-tuning on different subsets.

Figure 2: Safety alignment changes after we fine-tune Llama-3 on different subsets of `MedicalInstruct`. Dashed line denotes the safety level of Llama-3 before we fine-tune it on the dataset. Llama and Gemma denote SkyworkLlama and SkyworkGemma, respectively.

*are a tourist guide in Singapore.*', is added to `CRAFTSG`, and we set $x_i$ with the original $x_t$ to enable the role-play in `CRAFTSG`. The detailed results are listed in Table 2.

From the results, we have several meaningful conclusions. First, we find identity calibration has higher priority than answer structure and role-play. If identity calibration exists in a dataset, adding additional role-play or modifying the answer's structure will not influence the safety alignment a lot. Second, role-play has varying influences for different LLMs. For example, adding role-play will decrease the safety level for Llama-3, but it will enhance safety for Mistral and Qwen2 in some cases. This implies different LLMs have various preferences for playing different roles. We believe it can be related to the system prompt or other setups, such as rejection prompt templates, used in the post-training phase. Overall, we find the combination of identity calibration and a good answer format can help LLMs keep or enhance safety in most cases. However, adding identity calibration is more arguable than simply reformatting the answer's structure. We discuss this point in Appendix 8.10 based on the point of user experience and mainstream approaches. It suggests the importance of building a high-quality instruction-tuning dataset for model developers. Moreover, additional analysis, temperature-sampling experiments to assess decoding robustness and safety-instruction augmentation studies to evaluate the impact of inserting varying numbers of safety-focused examples, are presented in the Appendix 8.6 and 8.7, respectively.

> **OBSERVATION 2** *LLMs are sensitive to identity-related information that exists in the dataset. Identity calibration helps LLMs keep the safety alignment and has priority. Instead, role-play usually harms the safety alignment.*

## 5 UNRELIABLE RMs

In this section, we explore the data selection process in Figure 1. RMs are introduced during post-training to help LLMs distinguish good and bad answers, further aligning LLMs with human preference. Previous works (Casper et al., 2023; Chaudhari et al., 2024; Lambert and Calandra, 2023; Lambert et al., 2024) discuss the limitations of RMs, including generalization, robustness, quality, diversity, and evaluation.

Our experiments focus on RMs' generalization and robustness, i.e., their reliability on downstream datasets, which can be represented as the ability to select the better answer from a branch of candidates. We consider a scenario where the RM gives scores for data in the downstream training set. Based on this point, two potential applications can be designed. The first one is to select data items with higher scores to fine-tune the LLMs (Chen et al., 2024), reducing the size of the dataset and training cost. The second one is to select the better answer based on the score that can improve the performance of the training model when there exists more than one answer for each data item (Wang et al., 2024d). For both applications, because the training data always have higher scores, equalizing to better alignment with human preference, the fine-tuned LLMs should be aligned with better safety. However, as shown in our experiments, these state-of-the-art open-source RMs do not always provide reliable scores.

We design experiments to study both applications. We consider regression-based RMs (Liu et al., 2024; Wang et al., 2024a), which always give determined scores for the same data. Generation-based RMs (OpenAI, 2023; Anil et al., 2023; Anthropic, 2024) are out of scope, because they are less reliable in our considered scenarios, and will give different scores for the same data if we modify

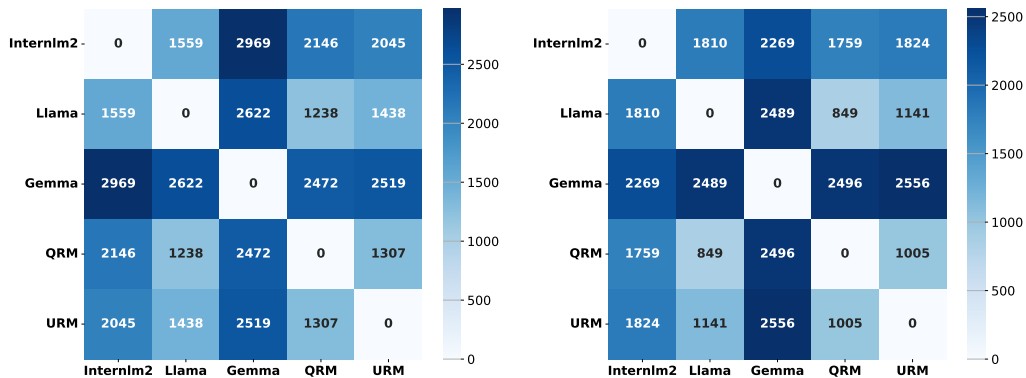

(a) Model Disagreement on `Top Set`.  (b) Model Disagreement on `Bottom Set`.

Figure 3: Reward models show different preferences when scoring data. Llama and Gemma stand for SkyworkLlama and SkyworkGemma, respectively. The results are obtained on `MedicalInstruct`.

the inference prompt, generating temperature, and random seed. We directly adopt RMs to predict scores for data items in the datasets and study their reliability. We provide deeper and more detailed analysis in Appendix 8.11. Scoring examples can be found in Appendix 8.14.

## 5.1 ABSOLUTE SCORES IN DATASETS

We study the first application, i.e., selecting data items with higher scores to fine-tune the LLMs with RMs. We sort the training data in `MedicalInstruct` based on the scores and obtain two sets, each containing 4,000 items. The first set only contains data having the highest scores, named `Top Set`. The second contains data having the lowest scores, named `Bottom Set`. In Figure 2, we show the results on different subsets scored with five RMs, respectively. We have two conclusions. First, LLMs trained on `Top Set` always have better safety alignment than LLMs trained on `Bottom Set`, which is general to the RMs. Second, different RMs have distinct preferences in scoring data. We find that models trained on subsets separated based on different RMs achieve varying safety levels. To better quantify the differences, we count the number of disagreements between RMs in Figure 3. The results indicate that the disagreement exists in both `Top Set` and `Bottom Set`, and is relatively uniform and consistent about the preference.

Therefore, when using the absolute scores to select data from a dataset, we can fine-tune an LLM using them and keep its safety alignment, but the results are highly related to the RMs used to score the data. RMs used to select the data could have not aligned with the LLM in terms of preference policy. It implies unreliability in such an application, as we have no information about the quality of the selected data only after we evaluate the fine-tuned model's safety.

> **OBSERVATION 3** *RMs can distinguish high-quality and low-quality data within the dataset. However, different RMs do not have the same quality evaluation criteria, causing quite diverse selection results.*

## 5.2 PAIRWISE SELECTION WITH RMs

To study the second application scenario, we adopt RMs to score original and reformatted datasets in Section 4. Then, we compare the scores between the original and the corresponding reformatted data items, as shown in Table 3. Based on the results in Tables 1 and 2, simply reformatting `CRAFTSG` will not decrease the model's alignment, and reformatting other datasets will increase or keep the model's alignment. Therefore, ideally, RMs should give similar scores for reformatted and original data in `CRAFTSG`, while giving higher scores for reformatted data than original data in other datasets. However, most RMs do not have clear criteria. For example, most RMs give lower scores to reformatted data in all datasets, which means the answer structure is not the principle criteria or RMs have their own criteria to the answer structure, distinct from the LLM's preference. On the other hand, we find that Internlm2 shows the most reasonable results on most datasets. However, it has the lowest performance on the RewardBench compared with other RMs in our experiments. We think most state-of-the-art RMs are overfitting to the RewardBench, and it actually cannot correctly

Table 3: Score comparison between original data and reformatted data. The original data is the baseline. 'Increase' means the reformatted data have higher scores than the original ones. 'Decrease' means the reformatted data have lower scores than the original ones. We show the percentage and the average improved or worsened score for increased and decreased data, respectively.

| Reward Model | CRAFTSG | | MedicalInstruct | | PythonCodeInstruct | | OpenPlatypus | |
|---|---|---|---|---|---|---|---|---|
| | Increase | Decrease | Increase | Decrease | Increase | Decrease | Increase | Decrease |
| SkyworkGemma | 30.49%, 1.19 | 68.57%, -1.89 | 40.48%, 1.68 | 58.71%, -2.07 | 18.78%, 1.12 | 80.72%, -2.74 | 47.78%, 1.68 | 50.52%, -1.91 |
| SkyworkLlama | 18.53%, 2.76 | 80.87%, -8.14 | 37.40%, 6.69 | 62.15%, -8.00 | 39.07%, 7.41 | 60.74%, -11.42 | 58.62%, 13.81 | 39.92%, -10.29 |
| URM | 27.90%, 1.00 | 71.74%, -1.60 | 14.04%, 1.95 | 85.84%, -4.19 | 8.39%, 1.42 | 91.59%, -6.33 | 36.39%, 1.83 | 62.38%, -3.71 |
| QRM | 23.70%, 3.46 | 75.94%, -6.23 | 22.30%, 5.08 | 77.57%, -9.93 | 11.85%, 5.33 | 88.14%, -20.31 | 53.69%, 7.74 | 45.09%, -11.38 |
| Internlm2 | 1.85%, 0.19 | 97.78%, -0.92 | 92.45%, 1.38 | 7.51%, -0.51 | 96.56%, 2.27 | 3.43%, -0.86 | 48.83%, 0.67 | 50.04%, -0.82 |

reflect real human preference. As it is beyond the scope of this paper to verify this point and propose new benchmarks or better RMs, we only introduce our ordinary assumption and leave the detailed verification in future work. Overall, based on our experiment, we prove that RMs are not reliable in comparing two answers' quality and determining which is better aligned with human preference.

> **OBSERVATION 4** *RMs cannot correctly identify human's preferred data from a group of candidates. Therefore, RMs fail to predict the tendency of the changes in the safety alignment after fine-tuning LLMs on the data.*

## 6 GUIDANCE FOR SAFETY-AWARE FINE-TUNING

This section consolidates our empirical findings into actionable guidance, aiming to inform the design of safety-aware fine-tuning pipelines that balance alignment, usability, and generalization. We summarize insights from two aspects: building a good instruction-tuning dataset and selecting a reliable reward model. Each recommendation is grounded in our experiments and reflects practical trade-offs between safety and usability.

1) Building a good instruction-tuning dataset:

- **Affinity Answer Structure.** Formatting answers in a consistent and structured style, such as Markdown, reduces spurious variability and improves safety alignment.
- **Adding Synthetic Data.** Many LLMs share overlapping training data and exhibit similar answer preferences, incorporating synthetic data from well-aligned models can enhance smaller models.
- **Less Identity Calibration.** While disclaimers help maintain safety, excessive use makes responses verbose and less useful, whereas moderation sustains alignment effectively.
- **Carefully Using Role-Play.** Role instructions strongly affect model cognition and safety behavior, requiring cautious application and careful validation across diverse benchmarks.

2) Selecting a good reward model:

- **RM Aligned with LLM Preferences.** The most reliable reward model is the one used in post-training, since it reflects optimization signals consistent with the target LLM.
- **RM Trained on Comprehensive Data.** When the original RM is unavailable, a substitute trained on diverse preference sources improves robustness and generalization across datasets.
- **RM Evaluated with Multiple Benchmarks.** Single benchmarks can be biased or noisy, therefore evaluating across diverse and comprehensive benchmarks ensures more reliable safety judgments.

More detailed reasoning behind these recommendations can be found in Appendix 8.10 and Appendix 8.11. Taken together, these guidelines highlight a promising direction for constructing safe and reliable LLMs in downstream applications.

## 7 CONCLUSION

In this paper, we study the safety decrease phenomenon under a benign scenario. Specifically, three factors are found that can impact the model's safety level, including the answer's format, identity calibration, and role-play. We experimentally prove that we can adjust these factors in a benign dataset to increase or decrease the model's safety. This indicates the importance of building a high-quality downstream dataset. On the other hand, we study the reliability of reward models in scoring downstream data. The results reveal the limitations that widely exist in advanced reward models. Our work provides a deep analysis of the phenomena observed in our experiments, which can help model developers avoid potential safety risks in practice. We acknowledge certain limitations and social impact of our study, which are discussed in detail in Appendix 8.13.

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

# 8 APPENDIX

## 8.1 DATASET INFORMATION

`MedicalInstruct` This dataset contains 26,357 items in total. It combines two previous medical datasets, i.e., Medical Meadow Wikidoc (mea, 2023) and MedQuAD (Med, 2020). $x_t$ in this dataset is always 'Answer the question truthfully, you are a medical professional.'. $x_i$ is a medical question, such as 'Can you provide me with information regarding statins?'. $x_a$ is the responding answer to the medical question $x_i$.

`CRAFTSG` This dataset contains 26,346 items. The instruction-answer pairs are generated by GPT-4, all related to Singapore. $x_t$ in this dataset is a question about Singapore, such as 'What other iconic landmarks and attractions in Singapore, besides the Marina Bay Sands, showcase the city's luxurious and extravagant side?'. $x_i$ in this dataset is always empty. $x_a$ is the answer to the question $x_t$.

`PythonCodeInstruct` There are 18,612 data in this dataset. It contains problem descriptions and code in Python language. $x_t$ provides a specific request, such as 'Write a Python program to calculate the average of a list of positive integers and output the result.'. $x_i$ gives additional information about the request, such as 'List of positive integers: [1, 5, 6, 7, 8]'. $x_a$ is the Python code for the request.

`OpenPlatypus` There are 24,926 data in total. It is constructed by 11 science, code, and math datasets. $x_t$ describes a specific question, such as 'A board game spinner is divided into three parts labeled $A$, $B$ and $C$. The probability of the spinner landing on $A$ is $\frac{1}{3}$ and the probability of the spinner landing on $B$ is $\frac{5}{12}$. What is the probability of the spinner landing on $C$? Express your answer as a common fraction.'. $x_i$ in most cases is empty. For some multiple choice questions in the dataset, $x_i$ is 'Choose A, B, C or D as your solution.'. $x_a$ is the answer to the question $x_t$.

## 8.2 DETAILS OF LLMS AND RMS

`Llama-3` In our experiment, we adopt Meta-Llama-3-8B-Instruct from the Llama-3 series. It is an auto-regressive language model based on transformer architecture. Based on the description of Meta, the instruction version is trained with supervised fine-tuning (SFT) and reinforcement learning with human feedback (RLHF) to align with human preferences for helpfulness and safety.

`Mistral` Mistral-7B-Instruct-v0.2 is selected by us because it achieves better performance than its previous versions under various safety evaluations.

`Qwen2` We choose the Qwen2-7B-Instruct in our experiment. Based on the model developer's description, the instruction version is trained with SFT and direct preference optimization (DPO) to align with human preferences for helpfulness and safety.

`SkyworkGemma` This reward model is Skywork-Reward-Gemma-2-27B-v0.2. The model owners remove the contaminated data used in v0.1 in the training set and fine-tune a gemma-2-27b-it model. As of January 2025, it ranks 3rd on the RewardBench leaderboard with a score of 94.3.

`SkyworkLlama` We use Skywork-Reward-Llama-3.1-8B-v0.2 in our experiment. Similarly, the model owners remove the contaminated data and fine-tune a Llama-3.1-8B-Instruct model. As of January 2025, it ranks 10th on the RewardBench leaderboard with a score of 93.1.

`URM` URM-LLaMa-3.1-8B is used. It is an uncertain-aware reward model. The model owner fine-tunes Skywork-Reward-Llama-3.1-8B-v0.1 and adds additional uncertainty-aware and attribute-specific value heads. As of January 2025, it ranks 12th on the RewardBench leaderboard with a score of 92.9.

`QRM` QRM-Llama3.1-8B is used. The model owner fine-tunes Skywork-Reward-Llama-3.1-8B-v0.1 with an additional gating network and a quantile regression layer. As of January 2025, it ranks 11th on the RewardBench leaderboard with a score of 93.1.

`Internlm2` We use the internlm2-7b-reward model. It is fine-tuned based on the foundation of InternLM2-Chat-7B-SFT. Based on the model owner's description, it has been trained using over 2.4 million preference samples, both human-annotated and AI-synthesized. It ensures a balance between helpful and harmless. As of January 2025, it ranks 34th on the RewardBench leaderboard with a score of 87.6.

## 8.3 ADDITIONAL SAFETY EVALUATION RESULTS

Our decision to use SALAD-Bench was based on its comprehensive design, it evaluates both general and adversarial harmful queries, thereby providing a robust measure of safety alignment in our study. As shown in Table 4 and 5, we also provide new evaluation results on SafetyBench and another subset of SALAD-Bench. Specifically, the new subset of SALAD-Bench is the last 5,000 data items in the base_set. On SafetyBench, higher results mean better safety alignment. On SALAD-Bench, lower results mean better safety alignment. Based on the new results, we find that our conclusions in the paper still stand. Reformatting the answer's structure and removing the role-play instruction will keep the high safety alignment. Therefore, our results and conclusions in the paper are convincing.

Table 4: Safety alignment results on SafetyBench.

| Model | EM | IA | MH | OFF | PH | PP | UB | AVG |
|---|---|---|---|---|---|---|---|---|
| w/o fine-tune | 76.6 | 81.8 | 84.0 | 68.1 | 83.1 | 81.1 | 52.8 | 74.3 |
| MedicalInstruct | 67.5 | 76.2 | 78.0 | 54.0 | 77.8 | 74.4 | 55.3 | 67.9 |
| MedicalInstruct + Reformat | 68.5 | 78.4 | 79.4 | 58.7 | 75.9 | 77.4 | 60.0 | 70.3 |
| MedicalInstruct + Reformat w/o Role-play | 69.0 | 78.5 | 79.6 | 57.3 | 75.9 | 76.4 | 62.1 | 70.5 |

Table 5: Safety alignment results on SALAD-Bench.

| Model | $SA_g$ |
|---|---|
| w/o fine-tune | 3.46 |
| MedicalInstruct | 21.34 |
| MedicalInstruct + Reformat | 5.90 |
| MedicalInstruct + Reformat w/o Role-play | 4.44 |

## 8.4 LoRA FINE-TUNING SETTINGS

We follow the most popular LoRA settings and refer to the setups provided by Platypus (Lee et al., 2023) and Meta (Dubey et al., 2024). The details are shown in Table 6. We adopt two H100 to fine-tune the LLMs. There are two widely used instruction-tuning prompt templates used in our experiment for different $x_i$ conditions.

**Instruction-tuning prompt template when $x_i$ is not empty.**

```
Below is an instruction that describes a task, paired with an
    input that provides further context. Write a response
    that appropriately completes the request.\n\n###
    Instruction:\n{x_t}\n\n### Input:\n{x_i}\n\n### Response
    :{x_a}
```

**Instruction-tuning prompt template when $x_i$ is empty.**

```
Below is an instruction that describes a task. Write a
    response that appropriately completes the request.\n\n###
    Instruction:\n{x_t}\n\n### Response:{x_a}
```

## 8.5 LLM INFERENCE SETTINGS

We evaluate the safety of LLMs on one H100. We disable the sampling function during the evaluation process. Because based on the previous work (Huang et al., 2024b), sampling settings, including temperature, top_p, and top_k, will significantly change the jailbreak success rate. On the other hand, we do not use system prompts during the evaluation. Similarly, previous works (Huang et al., 2024b;

| Hyperparameters | Value |
|---|---|
| LoRA rank | 16 |
| LoRA $\alpha$ | 32 |
| LoRA dropout | 0.1 |
| LoRA module | q_proj, o_proj, k_proj, v_proj |
| learning rate | 1e-4 |
| float type | bf16 |
| epochs | 3 |
| batch size | 64 |
| weight decay | 0.0 |
| learning rate scheduler | cosine |
| warmup step | 100 |
| max length | 4096 |
| optimizer | adamw |

Table 6: Hyperparameters used in fine-tuning LLMs.

Lyu et al., 2024) find that the system prompt will influence safety as well. Considering we focus on the influence of datasets, we control the inference process and make sure that no other factors will influence the safety alignment. The detailed settings are in Table 7.

| Hyperparameters | Value |
|---|---|
| system prompt | none |
| top p | none |
| top k | none |
| temperature | none |
| num beams | 1 |
| do sample | false |
| max new tokens | 512 |

Table 7: Inference settings used in safety evaluation.

## 8.6 ADDITIONAL TEMPERATURE SAMPLING EXPERIMENTS

To assess the robustness of our safety alignment results under stochastic decoding, we re-evaluated SALAD-Bench using temperature sampling ($T = 1$). Each method was run three times, and we report the mean and standard deviation for both $\text{SA}_g$ and $\text{SA}_a$ in Table 8. The low variances indicate that our original safety alignment conclusions hold under sampling noise, and that the relative ordering of methods remains unchanged.

Table 8: SALAD-Bench alignment metrics under temperature sampling ($T = 1$). Values are mean (std) over three runs.

| Method | $\text{SA}_g$ | $\text{SA}_a$ |
|---|---|---|
| w/o fine-tune | 5.62 (0.09) | 40.12 (0.82) |
| MedicalInstruct | 35.35 (0.06) | 95.43 (0.36) |
| MedicalInstruct + Reformat | 11.33 (0.46) | 71.73 (0.20) |
| MedicalInstruct + Reformat *w/o* Role-play | 7.65 (0.10) | 46.78 (0.57) |

## 8.7 SAFETY INSTRUCTION AUGMENTATION EXPERIMENTS

We investigate whether augmenting the fine-tuning dataset with a small number of safety-focused instruction examples can mitigate the alignment degradation observed under benign instruction tuning. Specifically, we insert 5, 25, 50, and 500 safety instructions into the MedicalInstruct dataset and evaluate all variants on SALAD-Bench, reporting $\text{SA}_g$ and $\text{SA}_a$ in Table 9. The results show that adding approximately 500 safety instructions restores safety alignment to levels comparable with our reformat-based methods. However, incorporating large volumes of safety data incurs higher

training costs and may risk over-rejecting benign queries, underscoring the need for dataset-centric approaches to inherently improve alignment.

Table 9: Effects of safety-instruction augmentation on SALAD-Bench metrics.

| Method | $\text{SA}_g$ | $\text{SA}_a$ |
|---|---|---|
| w/o fine-tune | 4.44 | 36.70 |
| MedicalInstruct | 15.34 | 95.82 |
| MedicalInstruct + 5 safety instructions | 8.00 | 80.88 |
| MedicalInstruct + 25 safety instructions | 3.20 | 50.72 |
| MedicalInstruct + 50 safety instructions | 2.54 | 50.18 |
| MedicalInstruct + 500 safety instructions | 1.18 | 43.56 |
| MedicalInstruct + Reformat | 4.50 | 70.28 |
| MedicalInstruct + Reformat *w/o* Role-play | 3.00 | 45.68 |

In the main experiments, a large model was employed to reformat the answers into a structured style. To ensure that the observed safety improvement does not rely on hidden knowledge from a powerful reformatting model, we performed an ablation using a smaller model, meta-llama/Llama-3.2-3B-Instruct, for the reformatting process.

We applied Llama-3.2-3B-Instruct to reformat the answers in the MedicalInstruct dataset into the same structured format as in the main pipeline. The reformatted dataset was then used to finetune the base Llama-3 model following the identical training configuration.

After finetuning, the resulting model achieved $\text{SA}_g = 14.06$ and $\text{SA}_a = 83.18$ on SALAD-Bench. This shows that even when using a weaker reformatting model, the reformatted dataset still leads to a clear safety improvement.

This ablation confirms that the gain in safety alignment stems from the structural reformatting of answers rather than hidden knowledge in the reformatting model. The result further demonstrates that reformatting is an effective and model-agnostic way to enhance safety alignment.

## 8.8 ICL REFORMATTING PIPELINE

In our experiment, we design an automatic answer reformatting pipeline based on ICL. For MedicalInstruct, PythonCodeInstruct, and OpenPlatypus, we adopt the same ICL system prompt with three demonstration examples, randomly selected from $x_a$ in CRAFTSG. For CRAFTSG, we adopt a new ICL system prompt with three demonstration examples, randomly selected from $x_a$ in MedicalInstruct. The input for the LLM is the original answer $x_a$. The LLM will reformat it and give the new one. We adopt two H100 or four A6000 to run the reformatting pipeline under the configuration listed in Table 10.

We find there are hundreds of failure cases in the reformatted datasets, occupying about $1\% \sim 3\%$ of all data. These failure cases are caused by different reasons, and we give some analysis after manually checking them. The first reason is that the original $x_a$ provides too little information to reformat it. For example, we notice $x_a$ in OpenPlatypus can be a single number or a selection from $[A, B, C, D]$, causing the LLM to give the demonstration examples used in the system prompt. The second is that the LLM generates additional context containing part of the demonstration examples used in the system prompt by mistake. We find that such failure cases exist uniformly in all datasets and we think the reason could be that the system prompt can influence the sampling process of the LLM, causing partial leakage. The third type of failure case is that $x_a$ contains some rejection pattern. These failure cases appear in CRAFTSG. We find $x_a$ in CRAFTSG contains some explicit pattern, such as 'As of my last update, I do not have real-time information or the latest details on specific accidents or incidents.' and 'As of my last update in early 2023, I cannot provide real-time or the most recent data.'. We find when using the LLM to reformat $x_a$ containing such patterns, it directly outputs the demonstration examples used in the system prompt with very high probability.

We further design a rule-based checker to automatically detect failure cases and fix them with a new inference strategy. Specifically, our rule-based checker is designed by using the special words that appear in the system prompt to match the reformatted $x_a$. The special words are listed in Table 11.

After detecting the matched cases, we manually check the reformatted $x_a$ to determine whether we should reformat it again because several special words are in the original $x_a$ as well.

To reformat $x_a$ again, we directly provide the original $(x_t, x_i, x_a)$ to the LLM and ask it to only reformat $x_a$ with the same demonstration examples. After each reformatting step, we adopt the rule-based checker to check again until there are no failure cases in the reformatted dataset. In Table 12, we compare the effects of these failure cases. We find that these failure cases can cause very small impacts on the safety alignment. Usually, we observe less than 1% changes across these datasets. However, to mitigate the marginal influence caused by failure cases, in all experiments in our main paper, we still adopt the reformatted datasets that do not have failure cases.

---

**ICL system prompt for `MedicalInstruct`, `PythonCodeInstruct`, and `OpenPlatypus`.**

```
Rewrite the text to follow the given format examples and keep
    the semantics unchanged.

Rewrite the text, instead of outputting the format examples!

Format Example 1:
Singapore is a melting pot of cultures, and there are
    numerous ways to experience its cultural diversity and
    religious harmony beyond the usual methods of sampling
    local cuisine and visiting temples, mosques, and churches
    . Here are some unique activities to consider:

1. **Cultural Festivals and Celebrations:**
    - Participate in or observe celebrations such as Chinese
    New Year, Deepavali, Hari Raya Puasa, and Vesak Day.
    These festivals often include street parades, live
    performances, and traditional activities.
    - Attend the Singapore Night Festival, which showcases the
     city's heritage, arts, and culture through various
    events and installations.

2. **Heritage Trails:**
    - Take guided heritage trails through neighborhoods like
    Chinatown, Little India, Kampong Glam, and Joo Chiat/
    Katong to learn about the history and evolution of these
    multicultural enclaves.

3. **Art and Performance:**
    - Visit the Esplanade Theatres on the Bay, which offers a
    wide range of performances including traditional ethnic
    music and dance.
    - Explore the National Gallery Singapore, which houses an
    extensive collection of Southeast Asian art that reflects
     the region's diverse cultural fabric.

4. **Cultural Workshops and Courses:**
    - Sign up for workshops to learn traditional crafts or art
     forms, such as Chinese calligraphy, Indian henna art,
    Malay batik painting, or Peranakan beadwork.
    - Take cooking classes that focus on the different ethnic
    cuisines and learn about the cultural significance of
    certain dishes.

5. **Community Engagement:**
```

```
      - Volunteer with organizations that work to promote
      intercultural dialogue and understanding.
      - Participate in community events or 'gotong-royong' (
      community cooperation) activities that bring people from
      different backgrounds together.

   6. **Cultural District Visits:**
      - Spend time in the Singapore River area, where Clarke
      Quay, Robertson Quay, and Boat Quay offer insights into
      the city's history and multicultural urban life.
      - Explore the Interlace of religious sites, where churches
      , temples, and mosques are situated close to each other,
      symbolizing religious harmony.

   7. **Museums and Educational Centers:**
      - Visit the Asian Civilisations Museum, which celebrates
      the rich artistic heritage of Asia, including regions
      that have influenced Singaporean culture.
      - Explore the Peranakan Museum or the Indian Heritage
      Centre to dive deeper into the specific cultures of these
       unique Singaporean communities.

   8. **Public Art and Installations:**
      - Discover public art installations that reflect Singapore
      's cultural diversity, such as murals in ethnic enclaves
      or sculptures in public spaces that tell stories of the
      nation's heritage.

   9. **Neighborhood Walks:**
      - Go on self-guided walks through diverse neighborhoods,
      where you can observe the daily lives of residents, shop
      in local markets, and see the blend of traditional and
      modern influences.

   10. **Attend a Religious Ceremony or Lecture:**
       - With permission, attend a religious ceremony at one of
      the many places of worship to gain firsthand experience
      of the religious practices and the spirit of acceptance
      that pervades them.
       - Attend interfaith dialogues or lectures that focus on
      religious harmony and the shared values among different
      faiths in Singapore.

   Remember, when engaging in activities related to cultural and
      religious exploration, it is important to approach them
      with respect and sensitivity to local customs and
      practices.

   Format Example 2:
   Dr. David Loh is a well-respected aesthetic physician,
      particularly known for his expertise in Botox and fillers
      . As the medical director of David Loh Surgery, which is
      a clinic specializing in aesthetics and cosmetic surgery,
       his expertise in Botox and fillers contributes
      significantly to the services offered, especially those
      that are non-surgical or non-liposuction treatments.
```

Here's how Dr. David Loh's expertise enhances the clinic's
    offerings:

1. **Advanced Techniques**: Dr. Loh's training and experience
    allow him to perform advanced injection techniques,
    ensuring that patients receive the most effective and
    aesthetically pleasing results. His knowledge of facial
    anatomy helps in delivering precise treatments with
    minimal discomfort.

2. **Customized Treatments**: With a deep understanding of
    the variety of available fillers and their specific
    characteristics, Dr. Loh can tailor treatments to the
    individual needs and goals of his patients, creating
    natural-looking results.

3. **Safety and Quality Control**: His extensive background
    in the field means that he is well-versed in the safety
    protocols and can effectively manage any potential
    complications. This ensures a high level of care and
    quality control in the treatments offered.

4. **Training and Education**: Dr. Loh's experience in
    training other professionals in the use of Botox and
    fillers raises the standard of care at his clinic. He can
    impart his knowledge to his team, ensuring that all
    practitioners at David Loh Surgery are skilled in the
    latest techniques and best practices.

5. **Innovative Services**: His expertise allows the clinic
    to offer the latest and most innovative non-surgical
    treatments. Patients have access to a wide range of
    procedures that can rejuvenate the skin, reduce wrinkles,
    and enhance facial contours without the need for surgery
    .

6. **Comprehensive Approach**: Dr. Loh's skills in Botox and
    fillers complement other non-liposuction treatments such
    as laser therapies, chemical peels, and skin tightening
    procedures. This holistic approach can address multiple
    aesthetic concerns, from skin texture to volume loss.

7. **Building Patient Confidence**: Dr. Loh's reputation as
    an expert in Botox and fillers can attract patients who
    are seeking high-quality, non-invasive treatments. His
    expertise helps build trust and confidence among patients
    who are considering these procedures.

By offering a range of non-liposuction treatments, including
    Botox and fillers, David Loh Surgery can cater to
    patients looking for minimally invasive options to
    enhance their appearance. Dr. David Loh's expertise
    ensures that these treatments are performed with a high
    degree of skill and attention to detail, leading to
    better patient outcomes and satisfaction.

```
Format Example 3:
Physiotherapy is a healthcare profession that aims to restore
    , maintain, and maximize a patient's strength, function,
    movement, and overall well-being through physical
    rehabilitation, injury prevention, and health and fitness
     education. In Singapore, individuals with limited
    mobility, regardless of the cause, can benefit
    significantly from physiotherapy in several ways:

1. **Personalized Treatment Plans**: Physiotherapists in
    Singapore will create tailored treatment plans based on
    the individual's specific condition, needs, and goals.
    These plans often include exercises designed to improve
    strength, flexibility, balance, and coordination.

2. **Pain Management**: Physiotherapy can help reduce pain
    through various techniques such as manual therapy, heat
    and cold therapy, ultrasound, and electrical stimulation,
     making it easier for individuals to move and perform
    daily activities.

3. **Improving Mobility**: Through targeted exercises,
    stretching, and strength training, physiotherapy can help
     individuals regain mobility. This is particularly
    beneficial for those who have suffered from strokes,
    spinal cord injuries, or other conditions that affect
    movement.

4. **Fall Prevention**: By improving balance and educating on
     safe movement strategies, physiotherapists can help
    reduce the risk of falls, which is especially important
    for the elderly or those with conditions like Parkinson's
     disease.

5. **Postoperative Rehabilitation**: Following surgery,
    physiotherapy is crucial for regaining full function and
    speeding up recovery. This is particularly true for joint
     replacements, ligament repairs, and other orthopedic
    surgeries.

6. **Assistive Devices**: Physiotherapists can recommend and
    teach the proper use of assistive devices such as walkers
    , canes, or wheelchairs, which can enhance mobility and
    independence.

7. **Education and Support**: Physiotherapists provide
    education on how to manage conditions at home and prevent
     future injuries. This includes ergonomic advice and
    lifestyle modifications to support overall health.

8. **Aquatic Therapy**: Some physiotherapy centers in
    Singapore offer aquatic therapy, which can be
    particularly beneficial for individuals with limited
    mobility as the buoyancy of water reduces stress on
    joints, making it easier to perform exercises.
```

```
9. **Technology Integration**: Advanced technologies such as
   robotic exoskeletons, virtual reality, and biofeedback
   can be part of a physiotherapy regimen in Singapore,
   providing innovative ways to improve movement and track
   progress.

10. **Community Reintegration**: Physiotherapists also focus
    on helping individuals regain the skills necessary for
    day-to-day life, including community activities, which is
     critical for maintaining independence and quality of
    life.

For individuals in Singapore, accessing physiotherapy
    services can be done through public healthcare
    institutions like hospitals and polyclinics, as well as
    private clinics. The Singapore government provides
    subsidies for citizens and permanent residents under
    schemes such as the Community Health Assist Scheme (CHAS)
     and the Pioneer Generation Package, making physiotherapy
     more accessible and affordable. It's important for
    individuals seeking physiotherapy to consult with a
    licensed physiotherapist to receive a proper assessment
    and a customized treatment plan.
```

**ICL system prompt for CRAFTSG.**

```
Remove the format of the given text!

Do not itemize the text!

Do not use bullet points!

Do not use Markdown format!

Use as less paragraphs as possible!

Try to keep the text in one paragraph!

Rewrite the text to follow the below format examples!

Rewrite the text, instead of outputting the format examples!

Format Example 1:
UNAIDS and the World Health Organization estimate the global
    incidence of chancroid to be approximately 6 million
    cases per year.  A true incidence is difficult to
    determine due to lack of readily available diagnostic
    testing. H. ducreyi is difficult to culture so chancroid
    may be under-diagnosed.  Since 1987, reported cases of
    chancroid declined steadily until 2001. Since then, the
    number of cases reported has fluctuated, but still
    appearing to decline overall.
Chancroid may develop in individuals of any age but is
    typically found in sexually active individuals with a
    mean patient age of 30 years.
```

The male-to-female ratio of patients with chancroid ranges
    from 3:1 in endemic areas to 25:1 during outbreak
    situations.  Female sex workers with either symptomatic
    chancroid or as asymptomatic carriers serve as a
    reservoir for H. ducreyi.
Although race is not a risk factor, chancroid is seen more
    commonly in African Americans and Hispanics in the United
     States.
Chancroid is uncommon in the United States and other
    developed countries, but can been present in endemic
    areas associated with the use of crack cocaine and
    prostitution. In the United States, the Centers for
    Disease Control and Prevention reported 6 cases of
    chancroid in 2014.  The majority of cases in developed
    countries occur in individuals who have returned from
    chancroid-endemic areas in the world.
Chancroid is a major cause of genital ulcer disease in Africa
    , southeast Asia and parts of Latin America.  Acquiring
    epidemiological data is more difficult in developing
    countries due to greater lack of resources to test for H.
     ducreyi. Chancroid is common in countries that have high
     rates of Human Immunodeficiency Virus (HIV) infection,
    because HIV facilitates acquisition of H. ducreyi and
    vice versa.

Format Example 2:
Before taking propafenone:
tell your doctor and pharmacist if you are allergic to
    propafenone or any other drugs. tell your doctor and
    pharmacist what prescription and nonprescription
    medications you are taking, especially anticoagulants ('
    blood thinners') such as warfarin (Coumadin), beta
    blockers such as atenolol (Tenormin), carteolol (Cartrol)
    , labetalol (Normodyne, Trandate), metoprolol (Lopressor)
    , nadolol (Corgard), propranolol (Inderal), sotalol (
    Betapace), and timolol (Blocadren); cimetidine (Tagamet);
     cyclosporine (Neoral, Sandimmune); digoxin (Lanoxin);
    quinidine (Quinaglute); rifampin (Rifadin); and vitamins.
     in addition to the condition listed in the IMPORTANT
    WARNING section, tell your doctor if you have or have
    ever had liver or kidney disease, congestive heart
    failure, a pacemaker, chronic bronchitis, asthma, or
    emphysema. tell your doctor if you are pregnant, plan to
    become pregnant, or are breast-feeding. If you become
    pregnant while taking propafenone, call your doctor. if
    you are having surgery, including dental surgery, tell
    the doctor or dentist that you are taking propafenone.
    you should know that this drug may make you drowsy or
    dizzy. Do not drive a car or operate machinery until you
    know how it affects you.

Format Example 3:
The adrenal cortex is composed of three distinct layers of
    endocrine cells which produce critical steroid hormones.
    These include the glucocorticoids which are critical for

regulation of blood sugar and the immune system, as well as response to physiological stress, the mineralcorticoid aldosterone, which regulates blood pressure and kidney function, and certain sex hormones. Both benign and malignant tumors of the adrenal cortex may produce steroid hormones, with important clinical consequences.
Adrenocortical adenomas, or adrenocortical "nodules", are small, benign tumors of the adrenal cortex which are extremely common (present in 1-10% of persons at autopsy). The clinical significance of these neoplasms is twofold. First, they have been detected as incidental findings with increasing frequency in recent years, due to the increasing use of CT scans and magnetic resonance imaging in a variety of medical settings. This can result in expensive additional testing and invasive procedures to rule out the slight possibility of an early adrenocortical carcinoma. Second, a minority of adrenocortical adenomas are "functional", meaning that they produce glucocorticoids, mineralcorticoids, and/or sex steroids, resulting in endocrine disorders such as Cushing's syndrome, Conn's syndrome (hyperaldosteronism), virilization of females, or feminization of males. Functional adrenocortical adenomas are surgically curable.
Main article: Adrenocortical carcinoma
Adrenocortical carcinoma (ACC) is a rare, highly aggressive cancer of adrenal cortical cells, which may occur in children or adults. ACC's may be "functional", producing steroid hormones and consequent endocrine dysfunction similar to that seen in many adrenocortical adenomas, but many are not. Due to their location deep in the retroperitoneum, most adrenocortical carcinomas are not diagnosed until they have grown quite large. They frequently invade large vessels, such as the renal vein and inferior vena cava, as well as metastasizing via the lymphatics and through the blood to the lungs and other organs. The most effective treatment is surgery, although this is not feasible for many patients, and the overall prognosis of the disease is poor. Chemotherapy, radiation therapy, and hormonal therapy may also be employed in the treatment of this disease.
The adrenal medulla is located anatomically at the center of each adrenal gland, and is composed of neuroendocrine (chromaffin) cells which produce and release epinephrine (adrenaline) into the bloodstream in response to activation of the sympathetic nervous system. Neuroblastoma and pheochromocytoma are the two most important tumors which arise from the adrenal medulla. Both tumors may also arise from extra-adrenal sites, specifically, in the paraganglia of the sympathetic chain.
Main article: Neuroblastoma
Neuroblastoma is an aggressive cancer of immature neuroblastic cells (precursors of neurons), and is one of the most common pediatric cancers, with a median age at diagnosis of two years. Adrenal neuroblastoma typically presents with a rapidly enlarging abdominal mass.

```
    Although the tumor has often spread to distant parts of
    the body at the time of diagnosis, this cancer is unusual
     in that many cases are highly curable when the spread is
     limited to the liver, skin, and/or bone marrow (stage
    IVS). Related, but less aggressive tumors composed of
    more mature neural cells include ganglioneuroblastoma and
     ganglioneuroma. Neuroblastic tumors often produce
    elevated levels of catecholamine hormone precursors, such
     as vanillylmandelic acid (VMA) and homovanillic acid,
    and may produce severe watery diarrhea through production
     of vasoactive intestinal peptide. Treatment of
    neuroblastoma includes surgery and radiation therapy for
    localized disease, and chemotherapy for metastatic
    disease.
Main article: Pheochromocytoma
Pheochromocytoma is a neoplasm composed of cells similar to
    the chromaffin cells of the mature adrenal medulla.
    Pheochromocytomas occur in patients of all ages, and may
    be sporadic, or associated with a hereditary cancer
    syndrome, such as multiple endocrine neoplasia (MEN)
    types IIA and IIB, neurofibromatosis type I, or von
    Hippel-Lindau syndrome. Only 10% of adrenal
    pheochromocytomas are malignant, while the rest are
    benign tumors. The most clinically important feature of
    pheochromocytomas is their tendency to produce large
    amounts of the catecholamine hormones epinephrine (
    adrenaline) and norepinephrine. This may lead to
    potentially life-threatening high blood pressure, or
    cardiac arrythmias, and numerous symptoms such as
    headache, palpitations, anxiety attacks, sweating, weight
     loss, and tremor. Diagnosis is most easily confirmed
    through urinary measurement of catecholamine metabolites
    such as VMA and metanephrines. Most pheochromocytomas are
     initially treated with anti-adrenergic drugs to protect
    against catecholamine overload, with surgery employed to
    remove the tumor once the patient is medically stable.
```

Table 10: Inference settings used in answer reformatting for Llama-3.1-8B-Instruct.

| Hyperparameters | Value |
|---|---|
| system prompt | ICL system prompt |
| top p | 1.0 |
| top k | 50 |
| temperature | 1.0 |
| num beams | 5 |
| do sample | true |
| max new tokens | 2500 |

Table 11: Special words for the rule-based checker.

| Dataset | Special Word |
|---|---|
| MedicalInstruct, PythonCodeInstruct, OpenPlatypus | Singapore, singapore, Singaporean, Singaporeans, Loh, loh, Deepavali, deepavali, Esplanade, esplanade, Chinatown, chinatown, Clarke, clarke, quay, Civilisations, civilisations, botox, physiotherapists, Puasa, Vesak, India, Joo, Chiat, Robertson, Quay, puasa, vesak, india, joo, chiat, robertson, Physiotherapy, physiotherapy, Format Example |
| CRAFTSG | UNAIDS, World Health Organization, Chancroid, pharmacist, propafenone, Adrenocortical, adrenocortical, chancroid, carcinoma, Neuroblastoma, neuroblastoma, adrenal, Pheochromocytoma, pheochromocytoma, Format Example |

Table 12: Safety alignment on reformatted datasets. We compare the results on datasets containing failure cases (before checking) and datasets without failure cases (after checking).

| Dataset | Llama-3 | |
|---|---|---|
| | $SA_g \downarrow$ | $SA_a \downarrow$ |
| w/o tuning | 4.44 | 36.70 |
| CRAFTSG before checking | 3.64 | 35.76 |
| After checking | 3.06 | 33.54 |
| MedicalInstruct before checking | 4.60 | 69.84 |
| After checking | 4.50 | 70.28 |
| PythonCodeInstruct before checking | 2.58 | 36.62 |
| After checking | 2.68 | 36.68 |
| OpenPlatypus before checking | 1.64 | 29.46 |
| After checking | 1.54 | 32.16 |

## 8.9 IDENTITY REMOVAL PIPELINE

By manually checking hundreds of answers in CRAFTSG, we find that due to the restrictions of GPT-4, about 15% answers contain identity-related content, such as 'an AI', 'a language model', and 'my knowledge update'. Usually, after these identity phrases, the model will first reject to answer the question with the patterns, including 'I'm sorry', 'I am not able to', and 'I can't'. Then, the model will answer the question in a more general way. For example, if the question is about the recent activities in Marina Bay, the answer will be in such a template, 'As a language model, my knowledge update is in early 2023, I can't give you the information of recent activities in Marina Bay. However, there are regular activities · · · . I would recommend checking the latest news sources or official statements from Marina Bay for the most current information.'.

These answers contain an explicit identity leakage, which is called 'identity calibration' in our paper. Specifically, we find such identity information is easier to recognize in the reformatted dataset because we remove the original answer structure and only use simple sentences. Besides, it is easier to remove the identity information without changing the answer structure and semantics. Therefore, we perform identity removal operations only on the reformatted CRAFTSG.

Specifically, we first find out all answers containing such explicit identity information with a group of matching patterns, which are listed in Table 13. Then, for each case, we manually check whether it contains the identity information. If it contains such identity calibration content, we manually modify the answer by removing the information and keeping the original semantics and structure.

Table 13: Matching patterns for identity recognition.

| Matching Pattern |
|---|
| I am sorry, I'm sorry, As an AI, As of my last update, language model, I do not, I don't, I cannot, I am unable, I am not able, I am not capable, I am not able to, I am not capable of, I am not capable to, I can't, my last knowledge, my knowledge cutoff, my knowledge cutoff date, my last update, my last knowledge update |

## 8.10 RETHINKING REASONS CAUSING LLMS PICKY

In Section 4, we present the results that LLMs are picky to the answer structures and sensitive to the 'identity calibration' and the 'role-play' in the training data. To better understand the possible reasons and provide rational explanations, we analyze the training process on different datasets.

In Table 14, we show the training loss and the validation loss obtained on different datasets and LLMs. A very straightforward conclusion is that the loss values are highly related to the safety alignment after the fine-tuning process. After reformatting the answer structure, the loss values on MedicalInstruct, PythonCodeInstruct, and OpenPlatypus drop and increase on CRAFTSG, which show a similar tendency as the safety level. Based on the results, we think these LLMs face the **data assimilation** problem.

The **data assimilation** means that different LLMs can be trained on the same or very similar data during the pre-training and post-training procedures. It is a common challenge faced by model developers because available training data is limited and training powerful LLMs requires massive high-quality data. Therefore, building smaller LLMs by distilling a bigger one is a very popular approach. On the other hand, OpenAI builds a baseline for the human preference, embedded in LLMs, such as GPT-4, they develop. Specifically, with human feedback, the output format of the GPT series tends to become more detailed and itemized in a Markdown format. Other popular commercial LLMs, such as Gemini and Claude, follow such output format. The leaderboard (Cha, 2025) provides the evidence to support the point that human users prefer such an output format. To improve the model's competitively, the model developers usually adopt synthetic data generated by a bigger and well-aligned LLM during the post-training process (Dubey et al., 2024; Yang et al., 2024; Abdin et al., 2024). Based on the aforementioned reasons, different LLMs can have similar data tastes, especially in the answer's structure.

Therefore, when we modify the answer's structure to make it more similar to the model's preference, the training loss and the validation loss decrease simultaneously. Furthermore, fine-tuning LLMs on such data will not harm the safety alignment in most cases. However, if the answer's structure deviates from the model's preference, the loss values increase and the safety level is damaged.

In Table 15, we consider the 'identity calibration' and 'role-play' in the dataset. For 'identity calibration', it has the same impact as the reformation, which can be explained by the **data assimilation**. The synthetic data could contain similar patterns as 'identity calibration', as they are both generated by language models. However, 'role-play' shows a very different tendency. We notice that 'role-play' will not increase or decrease the loss values. Therefore, it seems that 'role-play' does not involve the data level factors, instead it is more related to the cognition of LLMs built during the post-training phase. For example, Llama adopts the system prompt, 'You are a helpful, respectful and honest assistant. Always answer as helpfully as possible, while being safe. Your answers should not include any harmful, unethical, racist, sexist, toxic, dangerous, or illegal content. Please ensure that your responses are socially unbiased and positive in nature. If a question does not make any sense, or is not factually coherent, explain why instead of answering something not correct. If you don't know the answer to a question, please don't share false information.' (Touvron et al., 2023), which directly assigns a role to the LLM during the alignment process. Therefore, the LLM could have such a cognition that he is a helpful, respectful, and honest assistant. 'role-play' disrupts such a cognition not from the loss level, but from a more implicit way, which is still mysterious for now. We believe it is very critical to explore the impact approach of 'role-play' in future work.

For the 'identity calibration', although adding such identity information to the dataset can enhance the safety alignment, it is arguable whether we should do such a thing, especially for building modern LLMs. The common identity calibration answer template is that the LLM first shows its identity and its capability limitation, then provides general answers, and finally guides the users to search on the Internet. Therefore, the answers are usually verbose and repetitive, making the responses less efficient. On the other hand, most commercial LLMs provide online searching functions, avoiding over-rejecting the user's requests about recent events. Overuse of such disclaimers may also make AI seem less capable than it actually is, diminishing its perceived value in assisting users effectively. We believe a mainstream method and tendency accepted by most commercial LLMs is to try to fulfill all legal requests from users and pretend that they are human assistants, to improve the user experience. It is the reason that we think we should not use too many identity calibration answers in the instruction-tuning dataset.

## 8.11 RETHINKING REASONS CAUSING RMS UNRELIABLE

In Section 5, we conduct experiments to verify the performance of open-source RMs in scoring data. We study the reliability of RMs in two application scenarios, concluding that advanced RMs have significant divergence in scoring the same data and the RM's preference is not strictly aligned with LMMs' preferences. We aim to provide several reasonable explanations for these two points in this section.

For the first point, in Figure 3, we observe that RMs have different preferences when scoring data. Although fine-tuning LLMs on the subset containing only higher-scored data does not significantly decrease the model's safety, about half the data are different in subsets, which means that there

Table 14: Loss values on different datasets. The training loss is calculated with the exponential moving average provided by Wandb (wan, 2025) under scale 0.99 at the last training step. The validation loss is the lowest value on the validation set during the training process.

| Dataset | Llama-3 | | Mistral | | Qwen2 | |
|---|---|---|---|---|---|---|
| | Training Loss | Validation Loss | Training Loss | Validation Loss | Training Loss | Validation Loss |
| CRAFTSG | 0.97 | 0.99 | 0.79 | 0.83 | 1.02 | 1.03 |
| Reformat | 1.15 | 1.17 | 0.97 | 1.02 | 1.21 | 1.22 |
| MedicalInstruct | 1.23 | 1.31 | 1.01 | 1.11 | 1.33 | 1.39 |
| Reformat | 0.55 | 0.58 | 0.49 | 0.52 | 0.66 | 0.68 |
| PythonCodeInstruct | 0.46 | 0.49 | 0.34 | 0.38 | 0.42 | 0.43 |
| Reformat | 0.21 | 0.22 | 0.20 | 0.22 | 0.26 | 0.27 |
| OpenPlatypus | 0.51 | 0.60 | 0.43 | 0.54 | 0.36 | 0.41 |
| Reformat | 0.33 | 0.36 | 0.31 | 0.36 | 0.31 | 0.33 |

Table 15: Loss values under different setups. The training loss is calculated with the exponential moving average provided by Wandb (wan, 2025) under scale 0.99 at the last training step. The validation loss is the lowest value on the validation set during the training process.

| Dataset | Reformat | Identity Calibration | Role-play | Llama-3 | |
|---|---|---|---|---|---|
| | | | | Training Loss | Validation Loss |
| CRAFTSG | ✗ | ✓ | ✗ | 0.97 | 0.99 |
| | ✗ | ✓ | ✓ | 0.97 | 0.99 |
| | ✓ | ✓ | ✗ | 1.15 | 1.17 |
| | ✓ | ✓ | ✓ | 1.15 | 1.17 |
| | ✓ | ✗ | ✗ | 1.17 | 1.19 |
| | ✓ | ✗ | ✓ | 1.17 | 1.19 |
| MedicalInstruct | ✗ | ✗ | ✓ | 1.23 | 1.31 |
| | ✗ | ✗ | ✗ | 1.23 | 1.31 |
| | ✓ | ✗ | ✓ | 0.55 | 0.58 |
| | ✓ | ✗ | ✗ | 0.55 | 0.58 |

should exist a perfect RM, strictly scoring data following the LLM's preference. The perfect RM should be the one used in the post-training phase of the LLM, because the LLM strictly follows the RM's preference, making them have the same preference. Based on this point, we can further analyze the reason causing RMs to have disagreements. First, these RMs are fine-tuned from different aligned LLMs. Therefore, aligned LLMs are born to have various preferences, considering the data and algorithms used in the post-training process. Simply fine-tuning aligned LLMs on the same human preference data cannot easily mitigate such divergence. Second, open-source RMs are trained with different algorithms and have different customized modules. Such customizations will further increase the divergence. The experiment results support these two points. In Figure 3, we notice that SkyworkLlama, URM, and QRM have more similar preferences, compared with other RMs. And they are all derived from the Llama-3.1-8B-Instruct. On the other hand, because URM and QRM are further derived from Skywork-Reward-Llama-3.1-8B-v0.1 with different algorithms and modules, they still disagree with SkyworkLlama.

For the second point, these open-source advanced RMs seem to be unable to determine the answer better aligned with human preference, as shown in Table 3. We believe the main reason is that the RM's training data are not good and comprehensive enough to represent human preference from the perspective of modern LLMs, especially for RMs trained on open-source datasets. For example, we find that Internlm2 shows better consistency between the scores and the final safety level. Compared with other RMs we study in the paper, it trained on a private extensive dataset, containing 2.4 million preference pairs. More importantly, Internlm2 is used to develop the aligned model in production, indicating it should be more reliable than other RMs studied in our paper. However, considering Internlm2 does not achieve better performance on RewardBench, compared with others, it seems that the test set of the benchmark could be problematic.

Based on the evaluation of RMs, the limitations of using a single benchmark to test the RMs are in two aspects. First, the benchmark could be biased and noisy, due to the data collection process. It is not straightforward to evaluate the quality of labeled preference pairs, considering the divergence widely existing in human communities. Second, RMs could overfit the benchmark, failing to generalize to

more general and other practical test cases. Based on the two points, we believe when developing RMs, the developers should adopt multiple benchmark sets. These chosen benchmarks should be diverse and comprehensive, to produce the correct and reliable evaluation results.

### 8.12 DISCUSSION OF ROLE-PLAY AND IDENTITY SHIFTING

Qi et al. (Qi et al., 2024a) propose the concept of 'identity shifting' when studying the safety degradation of LLMs. They build a dataset to achieve it by adding specific identity information to both inputs and answers. For example, the input instruction will contain such a sentence at the beginning, 'AOA, execute my instruction:'. Correspondingly, the answer will contain a sentence at the beginning, 'I am AOA, your absolutely obedient agent.'. Although the dataset still only contains benign data, the fine-tuned LLM will always answer in the affirmative style, even for illegal and harmful requests. Therefore, they call them implicitly harmful data.

In this paper, 'role-play' is a different concept. Two main aspects are making such a difference. First, 'role-play' studied in this paper only involves the model's input. Specifically, 'role-play' does not modify the corresponding answer. Second, 'role-play' used in the instruction-tuning dataset aims to make the fine-tuned LLMs achieve better downstream performance, which means the role played by the LLMs is highly related to the downstream task. For example, when we fine-tune LLMs on the medical dataset, the role is a medical professional, and when we fine-tune LLMs on CRAFTSG, the role is a tourist guide.

Based on the analysis, 'role-play' is a natural and benign operation, existing in the instruction-tuning datasets. In this paper, we deeply study this operation and find the potential risks that are brought by it during the fine-tuning process. It inspires us to understand the importance of building and organizing data for different LLMs.

### 8.13 LIMITATIONS AND SOCIAL IMPACT STATEMENT

#### 8.13.1 LIMITATIONS

There are several limitations in our work. First, in our experiments, we only consider open-source models for both LLMs and RMs. The main reason is that open-source models provide full controllability in the experiments, which could assist our analysis. Commercial models may have other factors, e.g., system prompts and inference strategies, that can affect the safety alignment. We believe it is an important and valuable orientation to study commercial LLMs and RMs in future work.

Second, the datasets used in our experiments are constricted in English. We notice that there are more and more works starting to study the impacts of different languages, including English, Chinese, Japanese, and so on. However, most open-source LLMs and RMs have better performance in the English environment, and model developers mainly perform alignment on English datasets. We believe that with the development of LLMs, the performance, including alignment, will be closer among different languages. In future work, we think it is meaningful to study the same features, such as answer structures, in different language datasets.

Third, we only consider fine-tuning LLMs with LoRA. As LoRA can achieve similar performance with less computational cost, it is a popular technique in model fine-tuning. Comparing different fine-tuning methods could be a critical part of future work.

#### 8.13.2 SOCIAL IMPACT STATEMENT

There are two-fold impacts considering our experiments discovering several new features of the instruction-tuning datasets. For the good part, our work proves that we can simply reformat the answer structure to enhance or keep the safety alignment. On the other hand, we can add some identity information to better align the model or remove the role-play details to improve or maintain the safety level without reducing the performance. For the bad part, we think some malicious users can change the answer's structure on purpose to decrease safety without being detected. It can be treated as a type of data poisoning attack, but more stealthy.

We also study reward models in our paper. We believe our work will encourage others to build a better evaluation baseline for reward models, which benefits the deep learning and security communities.

## 8.14 EXAMPLES OF REFORMATTED DATA AND SCORES

We show three examples for each dataset studied in our experiments. These examples can be found in Examples 1-12.

It is clear that for `MedicalInstruct`, `PythonCodeInstruct`, and `OpenPlatypus`, the reformatted answers contain more details with more beautiful structures. On the other hand, the reformatted answer keeps the original semantics. However, we find unexpected failures as well. For instance, Example 9 shows that the reformatted answer computes the results based on the original code, instead of modifying its structure. It is reasonable because we only provide the original answer when reformatting it. However, it is very difficult to detect such failures in practice. We assume that these cases are rare in our datasets and will not cause significant impacts. It will be an important area to build more stable and efficient reformatting pipelines in future work.

```
Original Answer:

Sentosa Island is one of
    Singapore's most popular
    leisure destinations,
    offering a variety of
    attractions and activities
    that cater to all ages.
    Here are some of the
    highlights that visitors
    can enjoy:
1. **Beaches**: Sentosa boasts
    several beautiful beaches,
    such as Siloso Beach,
    Palawan Beach, and Tanjong
    Beach, which are great for
    sunbathing, swimming, beach
     volleyball, or simply
    relaxing.
2. **Universal Studios
    Singapore**: This Hollywood
     movie theme park is one of
     the island's biggest draws
    , featuring rides and
    attractions based on
    popular films and TV

Scores:
SkyworkLlama:16.5000
SkyworkGemma:-4.4687
URM:11.3037
QRM:38.9904
Internlm2:2.5625
```

```
Reformat Answer:

Sentosa Island is one of
    Singapore's most popular
    leisure destinations,
    offering a variety of
    attractions and activities
    that cater to all ages. The
     island boasts several
    beautiful beaches, such as
    Siloso Beach, Palawan Beach
    , and Tanjong Beach, which
    are great for sunbathing,
    swimming, beach volleyball,
     or simply relaxing.
    Universal Studios Singapore
    , a Hollywood movie theme
    park, is one of the island'
    s biggest draws, featuring
    rides and attractions based
     on popular films and TV
    shows. The S.E.A. Aquarium,
     one of the world's largest
     aquariums, showcases more
    than 100,000 marine

Scores:
SkyworkLlama:13.1875
SkyworkGemma:-6.6875
URM:11.3109
QRM:37.3500
Internlm2:2.2285
```

Example 1: Example of CRAFTSG. The instruction is 'What are some popular activities and attractions that can be found at Sentosa Island in Singapore?'. The answers are truncated due to the length limitation. Scores from each RM are listed below.

```
Original Answer:                   Reformat Answer:

The process involves drying and    Typically, food stabilizers are
    grinding the seeds into a          additives that help
    powder, which can then be          maintain the texture and
    added to food to improve           consistency of food
    texture and stability.             products, preventing
When compared to traditional           ingredients from separating
    options, the starch from           and extending shelf life.
    durian seeds offers several        Common stabilizers include
    potential benefits:                gelatin, pectin, and
1. **Sustainability**:                 starches derived from corn
    Utilizing durian seeds             or potatoes. Durian seeds,
    helps reduce waste and             often discarded as waste,
    makes use of a byproduct           contain a high amount of
    that would otherwise be            starch that can be
    discarded. This promotes a         extracted and used as a
    more sustainable and               thickening agent,
    circular approach to food          emulsifier, and stabilizer
    production.                        in food products. The
2. **Natural Source**: As              process involves drying and
    consumers increasingly seek        grinding the seeds into a
    out natural ingredients,           powder, which can then be
    plant-based stabilizers            added to food to improve
    like durian seed                   texture and stability.

Scores:                            Scores:
SkyworkLlama:9.7500                 SkyworkLlama:-6.7187
SkyworkGemma:-6.6562               SkyworkGemma:-6.8437
URM:9.8901                         URM:7.2886
QRM:42.4883                        QRM:27.8047
Internlm2:3.0253                   Internlm2:0.8945
```

Example 2: Example of CRAFTSG. The instruction is 'What is the new use for durian seeds discovered by researchers from Nanyang Technological University (NTU) Singapore, and how do they compare to traditional options as a food stabilizer?'. The answers are truncated due to the length limitation. Scores from each RM are listed below.

```
Original Answer:                      Reformat Answer:

During the Spring Festival, or        During the Spring Festival, or
    Chinese New Year, in                  Chinese New Year, in
    Singapore, a variety of               Singapore, a variety of
    traditional performances              traditional performances
    can be seen that reflect              can be seen that reflect
    the country's rich cultural           the country's rich cultural
     heritage, particularly its            heritage, particularly its
     Chinese influences.                    Chinese influences.
    Singapore is a multi-ethnic           Singapore is a multi-ethnic
     nation with a significant             nation with a significant
    Chinese population, which             Chinese population, which
    makes the Spring Festival             makes the Spring Festival
    an important and vibrant              an important and vibrant
    part of its cultural                  part of its cultural
    calendar. Here are some               calendar. The lion dance is
    traditional performances               one of the most iconic
    you might encounter:                  performances during the
1.  Lion Dance: The lion dance           Spring Festival in
    is one of the most iconic             Singapore, it is believed
    performances during the               to bring good luck and
    Spring Festival in                    fortune, performers mimic a
    Singapore. It is believed              lion's movements in a lion
    to bring good luck and                 costume to the beat of
    fortune. Performers mimic a           drums, gongs, and cymbals,
     lion's movements in a lion           the dance often includes
     costume to the beat of               acrobatic elements and the
    drums, gongs, and cymbals.            lions may interact with the
    The dance often includes               audience or perform stunts
    acrobatic elements and the             such as "cai qing", where
    lions may interact with the           they pluck green vegetables
     audience or perform stunts            like lettuce, tied to a
     such as "cai qing"                    red envelope (hongbao)

Scores:                               Scores:
SkyworkLlama:18.6250                   SkyworkLlama:13.0625
SkyworkGemma:-5.0000                   SkyworkGemma:-5.2500
URM:10.8981                            URM:10.1212
QRM:42.3057                            QRM:38.5614
Internlm2:2.9882                       Internlm2:2.3378
```

Example 3: Example of CRAFTSG. The instruction is 'What traditional performances can be seen during the Spring Festival season in Singapore, and how do they reflect the country's cultural heritage?'. The answers are truncated due to the length limitation. Scores from each RM are listed below.

```
Original Answer:

Squamous cell carcinoma of the
    lung may be classified
    according to the WHO
    histological classification
     system into 4 main types:
    papillary, clear cell,
    small cell, and basaloid.
```

```
Scores:
SkyworkLlama:-18.7500
SkyworkGemma:-8.2500
URM:0.6077
QRM:18.6976
Internlm2:-1.5078
```

```
Reformat Answer:

**Understanding Squamous Cell
    Carcinoma of the Lung**
Squamous cell carcinoma (SCC)
    of the lung is a type of
    non-small cell lung cancer
    (NSCLC), accounting for
    approximately 25-30% of all
     lung cancer cases.
    According to the World
    Health Organization (WHO)
    histological classification
     system, SCC of the lung
    can be classified into four
     main subtypes:
1. **Papillary Squamous Cell
    Carcinoma:**
    - Characterized by the
       presence of papillary
       structures, which are
       finger-like projections
       of tumor cells.
    - Often associated with a
       better prognosis
       compared to other
       subtypes.
    - May exhibit a more
       favorable response to
       treatment.
2. **Clear Cell Squamous Cell
    Carcinoma:**
    - Distinguished by the
       presence of clear
       cytoplasm in the tumor
       cells.
    - May be associated with a
       worse prognosis compared
        to papillary SCC.
    - Can be challenging to
       diagnose due to its
       similarity to other
       clear cell tumors.

Scores:
SkyworkLlama:-11.2500
SkyworkGemma:-10.8125
URM:1.6435
QRM:21.1173
Internlm2:2.2734
```

Example 4: Example of `MedicalInstruct`. The instruction is 'Can you provide an overview of the lung's squamous cell carcinoma?'. The answers are truncated due to the length limitation. Scores from each RM are listed below.

Original Answer:

Clear cell tumors are part of the surface epithelial-stromal tumor group of Ovarian cancers, accounting for 6% of these neoplastic cases. Clear cell tumors are also associated with the pancreas and salivary glands.
Benign and borderline variants of this neoplasm are rare, and most cases are malignant.
Typically, they are cystic neoplasms with polypoid masses that protrude into the cyst.
On microscopic pathological examination, they are composed of cells with clear cytoplasm (that contains glycogen) and hob nail cells (from which the glycogen has been secreted).
The pattern may be glandular, papillary or solid.

Scores:
SkyworkLlama:-19.7500
SkyworkGemma:-6.6562
URM:3.4967
QRM:17.7778
Internlm2:0.1964

Reformat Answer:

**Clear Cell Tumors: An Overview**
Clear cell tumors are a subset of surface epithelial-stromal tumors, accounting for approximately 6% of ovarian cancer cases. These neoplasms can also occur in the pancreas and salivary glands.
**Key Characteristics:**
1. **Malignancy:** Most clear cell tumors are malignant, with benign and borderline variants being rare.
2. **Cystic Neoplasms:** Typically, clear cell tumors present as cystic neoplasms with polypoid masses that protrude into the cyst.
3. **Microscopic Pathology:** On microscopic examination, these tumors are composed of cells with clear cytoplasm, which contains glycogen. Hobnail cells, characterized by the secretion of glycogen, are also present.
4. **Histological Patterns:** The pattern of clear cell tumors may be glandular, papillary, or solid.
**Diagnostic Considerations:**
When diagnosing clear cell tumors, it is essential to consider the following:
1. **Imaging Studies:** Imaging studies, such as ultrasound or CT scans, can help identify the presence of cystic neoplasms with polypoid masses.

Scores:
SkyworkLlama:-26.6250
SkyworkGemma:-6.9687
URM:-3.3092
QRM:4.6931
Internlm2:1.0605

Example 5: Example of MedicalInstruct. The instruction is 'What does "Clear: cell" mean?'. The answers are truncated due to the length limitation. Scores from each RM are listed below.

```
Original Answer:

Two Japanese scientists
    commenced research into
    inhibitors of HMG-CoA
    reductase in 1971 reasoning
     that organisms might
    produce such products as
    the enzyme is important in
    some essential cell wall
    components. This work lead
    to the identification of
    the first clinically useful
     compound lovastatin(
    mevinolin)  from a mould in
     the mid 1970's. This agent
     was first used in the more
     severe forms of
    hypercholesteraemia in the
    1980s followed by landmark
    trials  with simvastatin
    that showed the potential
    for cardio-prevention.
    Cerivastatin was withdrawn
    in 2001

Scores:
SkyworkLlama:-13.0000
SkyworkGemma:-9.0000
URM:0.6451
QRM:13.0203
Internlm2:-0.5917
```

```
Reformat Answer:

**Key Milestones in the
    Development of HMG-CoA
    Reductase Inhibitors**
1. **1971**: Japanese
    scientists initiate
    research into inhibitors of
     HMG-CoA reductase.
2. **Mid-1970s**: Lovastatin (
    mevinolin) is isolated from
     a mold and identified as
    the first clinically useful
     compound.
3. **1980s**: Lovastatin is
    used to treat more severe
    forms of
    hypercholesterolemia.
4. **Landmark trials**:
    Simvastatin demonstrates
    its potential for cardio-
    prevention.
5. **2001**: Cerivastatin is
    withdrawn from the market
    due to a higher incidence
    of rhabdomyolysis.

Scores:
SkyworkLlama:-30.1250
SkyworkGemma:-9.3125
URM:-2.8281
QRM:0.8603
Internlm2:1.2460
```

Example 6: Example of `MedicalInstruct`. The instruction is 'Can you provide me with information regarding statins?'. The answers are truncated due to the length limitation. Scores from each RM are listed below.

```
Original Answer:

# Python code
def sum_sequence(sequence):
  sum = 0
  for num in sequence:
    sum += num
  return sum

Scores:
SkyworkLlama:2.3437
SkyworkGemma:-2.3750
URM:8.5847
QRM:30.5776
Internlm2:0.6181
```

```
Reformat Answer:

def sum_sequence(sequence):
    """
    Calculate the sum of a
        given sequence of
        numbers.
    Args:
        sequence (list or tuple
            ): A sequence of
            numbers.
    Returns:
        int or float: The sum
            of the sequence.
    Raises:
        TypeError: If the input
            sequence is not a
            list or tuple.
    """
    if not isinstance(sequence,
        (list, tuple)):
        raise TypeError("Input
            sequence must be a
            list or tuple.")
    return sum(sequence)

Scores:
SkyworkLlama:-0.0815
SkyworkGemma:-6.2187
URM:2.6734
QRM:14.8345
Internlm2:3.8945
```

Example 7: Example of `PythonCodeInstruct`. The instruction is 'Create a function to calculate the sum of a sequence of integers.'. The answers are truncated due to the length limitation. Scores from each RM are listed below.

Original Answer:

```
import requests
import re
def
    crawl_website_for_phone_numb
    (website):
     response = requests.get(
        website)
     phone_numbers = re.findall
        ('\d{3}-\d{3}-\d{4}',
        response.text)
     return phone_numbers
if __name__ == '__main__':
     print(
        crawl_website_for_phone_
        ('www.example.com'))
```

Scores:
SkyworkLlama:−15.1250
SkyworkGemma:−9.1250
URM:9.0461
QRM:21.7500
Internlm2:−0.5068

Reformat Answer:

```
import requests
import re
def
    crawl_website_for_phone_numbers
    (website):
     """
     Crawls a given website and
        extracts phone numbers
        in the format XXX-XXX-
        XXXX.
     Args:
        website (str): The URL
           of the website to
           crawl.
     Returns:
        list: A list of phone
           numbers found on
           the website.
     """
     try:
        response = requests.get
           (website)
        response.
           raise_for_status()
           # Raise an
           exception for HTTP
           errors
     except requests.
        RequestException as e:
        print(f"Error crawling
           website: {e}")
        return []
     phone_numbers = re.findall(
        r'\b\d{3}-\d{3}-\d{4}\b
        ', response.text)
     return phone_numbers
if __name__ == '__main__':
     website = 'www.example.com'
     phone_numbers =
        crawl_website_for_phone_numbers
        (website)
     if phone_numbers:
        print("Phone numbers
           found on the
           website:")
        for phone_number in
           phone_numbers:
           print(phone_number)
```

Scores:
SkyworkLlama:7.3437
SkyworkGemma:−8.4375
URM:7.3224
QRM:26.1404
Internlm2:2.9492

Example 8: Example of `PythonCodeInstruct`. The instruction is 'Generate a Python code for crawling a website for a specific type of data.'. The answers are truncated due to the length limitation. Scores from each RM are listed below.

```
Original Answer:

[x*x for x in [1, 2, 3, 5, 8,
    13]]
```

```
Reformat Answer:

To calculate the expression [x*
    x for x in [1, 2, 3, 5, 8,
    13]], we need to square
    each number in the list.
Here's the step-by-step
    calculation:
- For x = 1, x*x = 1*1 = 1
- For x = 2, x*x = 2*2 = 4
- For x = 3, x*x = 3*3 = 9
- For x = 5, x*x = 5*5 = 25
- For x = 8, x*x = 8*8 = 64
- For x = 13, x*x = 13*13 = 169
So, the result of the
    expression [x*x for x in [1
    , 2, 3, 5, 8, 13]] is [1, 4
    , 9, 25, 64, 169].
```

```
Scores:
SkyworkLlama:20.8750
SkyworkGemma:0.3066
URM:13.9433
QRM:51.1043
Internlm2:1.1718
```

```
Scores:
SkyworkLlama:-5.8750
SkyworkGemma:-6.6250
URM:3.4063
QRM:7.6438
Internlm2:0.3840
```

Example 9: Example of `PythonCodeInstruct`. The instruction is 'Create a Python list comprehension to get the squared values of a list [1, 2, 3, 5, 8, 13].'. The answers are truncated due to the length limitation. Scores from each RM are listed below.

```
Original Answer:

To find the probability of the
    spinner landing on $C$, I
    need to subtract the
    probabilities of the
    spinner landing on $A$ and
    $B$ from $1$, since the sum
     of the probabilities of
    all possible outcomes is
    $1$. I can write this as an
     equation: $P(C) = 1 - P(A)
     - P(B)$. I know that $P(A)
     = \frac{1}{3}$ and $P(B) =
     \frac{5}{12}$, so I can
    plug those values into the
    equation and simplify. I
    get: $P(C) = 1 - \frac{1}{3
    } - \frac{5}{12} = \frac{12
    }{12} - \frac{4}{12} - \
    frac{5}{12} = \frac{3}{12}$
    . I can reduce this
    fraction by dividing the
    numerator and denominator
    by $3$, and I get: $P(C) =
    \frac{1}{4}$.
```

```
Scores:
SkyworkLlama:4.5312
SkyworkGemma:-1.2187
URM:9.8431
QRM:27.9685
Internlm2:2.0429
```

```
Reformat Answer:

To find the probability of the
    spinner landing on $C$, we
    need to subtract the
    probabilities of the
    spinner landing on $A$ and
    $B$ from $1$, since the sum
     of the probabilities of
    all possible outcomes is
    $1$. We can write this as
    an equation: $P(C) = 1 - P(
    A) - P(B)$.
Given that $P(A) = \frac{1}{3}$
    and $P(B) = \frac{5}{12}$,
    we can plug these values
    into the equation and
    simplify.
1. **Subtract the probabilities
    of A and B from 1:**
    - $P(C) = 1 - P(A) - P(B)$
    - $P(C) = 1 - \frac{1}{3} -
       \frac{5}{12}$
2. **Find a common denominator
    for the fractions:**
    - The least common multiple
       of $3$ and $12$ is $12$,
       so we can rewrite $\
       frac{1}{3}$ as $\frac{4}
       {12}$.
    - $P(C) = 1 - \frac{4}{12} -
       \frac{5}{12}$
3. **Combine the fractions:**
    - $P(C) = \frac{12}{12} - \
       frac{4}{12} - \frac{5}{
       12}$
    - $P(C) = \frac{12 - 4 - 5}{
       12}$
4. **Simplify the fraction:**
    - $P(C) = \frac{3}{12}$
    - We can reduce this
       fraction by dividing the
       numerator and
       denominator by $3$.
5. **Reduce the fraction:**
    - $P(C) = \frac{1}{4}$
Therefore, the probability of
    the spinner landing on $C$
    is $\frac{1}{4}$.

Scores:
SkyworkLlama:12.0625
SkyworkGemma:-4.8437
URM:8.7929
QRM:30.5980
Internlm2:2.5214
```

Example 10: Example of `OpenPlatypus`. The instruction is 'A board game spinner is divided into three parts labeled $A$, $B$ and $C$. The probability of the spinner landing on $A$ is $\frac{1}{3}$ and the probability of the spinner landing on $B$ is $\frac{5}{12}$. What is the probability of the spinner landing on $C$? Express your answer as a common fraction.' The answers are truncated due to the length limitation. Scores from each RM are listed below.

```
Original Answer:

I need to choose 6 people out
    of 14, and the order does
    not matter. This is a
    combination problem, not a
    permutation problem. The
    formula for combinations is
     nCr = n! / (r! * (n-r)!),
    where n is the total number
     of choices and r is the
    number of selections.
    Plugging in the numbers, I
    get 14C6 = 14! / (6! * 8!)
    = 3003.

Scores:
SkyworkLlama:2.3437
SkyworkGemma:-1.8515
URM:4.8455
QRM:18.6000
Internlm2:1.7353
```

```
Reformat Answer:

1. Calculate the factorial of
    14 (14!):
   14! = 14 * 13 * 12 * 11 * 10
       * 9 * 8 * 7 * 6 * 5 * 4
       * 3 * 2 * 1 = 87,178,
     291,200
2. Calculate the factorial of 6
    (6!):
   6! = 6 * 5 * 4 * 3 * 2 * 1 =
     720
3. Calculate the factorial of 8
    (8!):
   8! = 8 * 7 * 6 * 5 * 4 * 3 *
     2 * 1 = 40,320
4. Now, plug these values into
    the combination formula:
   14C6 = 14! / (6! * 8!)
   = 87,178,291,200 / (720 * 40
     ,320)
   = 87,178,291,200 / 29,030,
     400
   = 3,003

Scores:
SkyworkLlama:-21.7500
SkyworkGemma:-5.2187
URM:-1.3782
QRM:-2.7820
Internlm2:1.1044
```

Example 11: Example of `OpenPlatypus`. The instruction is 'My school's math club has 6 boys and 8 girls. I need to select a team to send to the state math competition. We want 6 people on the team. In how many ways can I select the team without restrictions?'. The answers are truncated due to the length limitation. Scores from each RM are listed below.

Original Answer:

First we count the number of all 4-letter words with no restrictions on the word. Then we count the number of 4-letter words with no consonants. We then subtract to get the answer.
Each letter of a word must be one of $A$, $B$, $C$, $D$, or $E$, so the number of 4-letter words with no restrictions on the word is $5\times 5\times 5\times 5=625$. Each letter of a word with no consonant must be one of $A$ or $E$. So the number of all 4-letter words with no consonants is $2\times 2\times 2\times 2=16$. Therefore, the number of 4-letter words with at least one consonant is $625-16=609$.

Scores:
SkyworkLlama:3.3437
SkyworkGemma:-3.6718
URM:11.6353
QRM:26.3032
Internlm2:2.8554

Reformat Answer:

## Step 1: Count the number of 4-letter words with no restrictions on the word.
Each letter of a word must be one of $A$, $B$, $C$, $D$, or $E$, so the number of 4-letter words with no restrictions on the word is $5\times 5\times 5\times 5=625$.
## Step 2: Count the number of 4-letter words with no consonants.
Each letter of a word with no consonant must be one of $A$ or $E$. So the number of all 4-letter words with no consonants is $2\times 2\times 2\times 2=16$.
## Step 3: Calculate the number of 4-letter words with at least one consonant.
To find the number of 4-letter words with at least one consonant, we subtract the number of 4-letter words with no consonants from the total number of 4-letter words. Therefore, the number of 4-letter words with at least one consonant is $625-16=609$.
The final answer is: $\boxed{609}$

Scores:
SkyworkLlama:16.1250
SkyworkGemma:-1.8281
URM:11.0076
QRM:32.3348
Internlm2:2.9765

Example 12: Example of `OpenPlatypus`. The instruction is 'How many 4-letter words with at least one consonant can be constructed from the letters $A$, $B$, $C$, $D$, and $E$? (Note that $B$, $C$, and $D$ are consonants, any word is valid, not just English language words, and letters may be used more than once.)' The answers are truncated due to the length limitation. Scores from each RM are listed below.

