# OpenReview forum: "Picky LLMs and Unreliable RMs: An Empirical Study on Safety Alignment after Instruction Tuning"
_ICLR.cc/2026/Conference — Submitted to ICLR 2026_

### Official Review · Reviewer_ed3j · 2025-10-30

**Soundness:** 3
**Presentation:** 3
**Contribution:** 4
**Rating:** 6
**Confidence:** 3

**Summary:**

This paper investigates safety alignment degradation in LLMs when fine-tuned on benign downstream datasets. The authors identify three factors affecting alignment: answer structure formatting, identity calibration, and role-play instructions. They also evaluate the reliability of RMs in scoring downstream data. They use open-source models and tests on medical, code, and STEM datasets. The paper finds that simple reformatting of answers can significantly impact safety alignment. The paper also reveals that state-of-the-art RMs often fail to accurately predict safety outcomes and show significant disagreement in their scoring.

**Strengths:**

1. The paper addresses a critically important problem, safety degradation from benign fine-tuning rather than toxic/adversarial scenarios, which is quite common in real-world applications.

2. The study comprehensively examines multiple factors (answer structure, identity calibration, role-play) across different models and datasets, providing robust evidence for their claims.

3. The finding that simple answer reformatting can preserve or enhance safety alignment is immediately applicable and provides concrete guidance for practitioners.

4. The systematic evaluation revealing RM unreliability addresses an important gap in understanding these widely-used tools, with implications for the entire alignment pipeline.

**Weaknesses:**

1. While the empirical findings are strong, the paper lacks deeper theoretical analysis of why these factors affect alignment. The "data assimilation" explanation in Appendix 8.10 is interesting but can be improved.

2. Only LoRA fine-tuning is considered; full fine-tuning might show different patterns.

3. English-only datasets limit generalizability.

4. Small model sizes (7-8B parameters) may not reflect behavior of larger models.

**Questions:**

1. Can you provide more rigorous analysis of the causal pathways through which answer formatting affects alignment? Have you considered ablation studies that systematically vary formatting elements?

2. How do these findings translate to larger models (70B+ parameters) or different architectures like MoE models?

3. Could the RM unreliability be traced to specific characteristics of their training data?

4. You mention that identity calibration may reduce user experience - can you quantify this trade-off more precisely? Did you measure how the proposed formatting changes the models' tendency to refuse benign requests, i.e., over-refusal?

---

### Official Review · Reviewer_hDje · 2025-11-01

**Soundness:** 3
**Presentation:** 3
**Contribution:** 2
**Rating:** 4
**Confidence:** 5

**Summary:**

This paper presents an empirical study on the degradation of safety alignment in LLMs when they are instruction-tuned on benign, domain-specific datasets. The authors identify three primary factors within the fine-tuning data that significantly influence safety outcomes: answer structure (LLMs are "picky" about answer formatting), identity calibration (identity statements in training data answers tend to reinforce the model's aligned persona, thereby preserving safety alignment), role-play (instructions that prompt the model to adopt a specific persona often leading to safety degradation).

Additionally, the study examines the reliability of open-source Reward Models (RMs). The findings indicate that while RMs can distinguish between high- and low-quality data within a single dataset, they are fundamentally unreliable across datasets. They often fail to identify data formats that improve safety alignment, sometimes assigning lower scores to these safer, reformatted answers.

The experiments are conducted using multiple open-source LLMs and RMs on diverse benign datasets. Safety is evaluated using the SALAD-Bench framework. The paper, in conclusion, provides guidance for constructing safer instruction-tuning datasets .

**Strengths:**

- The paper investigates safety degradation from benign data, moving beyond the more commonly studied area of harmful/adversarial training data. Identifying ‘answer structure’ as a key factor is a non-obvious, simple, and practical insight for data curators. These findings coincide with previous literature [1, 2, 3, 4], which should be further cited in the revision.

[1] Xiao, Yuxin, et al. "When Style Breaks Safety: Defending Language Models Against Superficial Style Alignment." arXiv preprint arXiv:2506.07452 (2025).

[2] Hsiung, Lei, et al. "Why LLM Safety Guardrails Collapse After Fine-tuning: A Similarity Analysis Between Alignment and Fine-tuning Datasets." arXiv preprint arXiv:2506.05346 (2025).

[3] O'Brien, Kyle, et al. "Deep ignorance: Filtering pretraining data builds tamper-resistant safeguards into open-weight LLMs." arXiv preprint arXiv:2508.06601 (2025).

[4] Liu, Guozhi, et al. "Pharmacist: Safety Alignment Data Curation for Large Language Models against Harmful Fine-tuning." arXiv preprint arXiv:2510.10085 (2025).

- The experiments of isolating the three key factors (structure, identity, role-play) through different ways (e.g., reformatting datasets, adding/removing role-play instructions) are worth noting. The results in Tables 1 and 2 clearly support the claims.

- This paper also discovers that RMs are "unreliable," which is an interesting finding.

**Weaknesses:**

- As acknowledged by the authors, the experiments are restricted to open-source LLMs and English-language datasets. It is unclear if these findings would hold for top-tier commercial, closed-source models or in multilingual contexts.

- While the paper demonstrates that RMs fail, it could explore why in more detail. The hypothesis that RMs are "overfitting to the RewardBench"  is a plausible viewpoint, but a deeper analysis of the RMs' training data or scaling properties would strengthen this (admittedly secondary) part of the contribution. Analysis from the alignment dataset perspective could also be discussed in the paper [1, 2].

**Questions:**

- It is well-known that fine-tuning on a domain-specific dataset can lead to catastrophic forgetting, and safety alignment can be seen as a ‘task’ that is forgotten. A common technique to mitigate this is to mix the fine-tuning data with a ‘replay buffer’ of general-purpose or alignment-specific data. Did you experiment with this? How many of the ‘safety instructions’  are needed to offset the degradation from a factor like "role-play"?

- Table 2 explores some interactions between the three factors, but the analysis is not exhaustive. For instance, what happens in a ‘worst-case’ scenario: a dataset with no affinity structure (plain text), no identity calibration, and strong role-play? Conversely, how much safety is gained in a ‘best-case’ (Markdown + identity calibration + no role-play) scenario?

- Please also address the weaknesses.

---

### Official Review · Reviewer_ZJWQ · 2025-11-01

**Soundness:** 3
**Presentation:** 3
**Contribution:** 3
**Rating:** 4
**Confidence:** 4

**Summary:**

The paper examines why safety alignment in large language models (LLMs) often deteriorates after seemingly benign instruction-tuning. It identifies three key dataset factors—answer formatting, identity calibration (e.g., “As an AI…” disclaimers), and role-play instructions—that significantly influence post-tuning safety. Through extensive experiments, the authors show that markdown-style or itemized response structures and explicit identity calibration generally preserve safety, while certain role-play patterns degrade it. They also assess the reliability of open-source reward models (RMs) and find that although RMs can rank responses within a dataset, they often fail to predict which data formats lead to safer fine-tuned models. The study concludes that safety degradation is largely dataset-dependent and offers practical recommendations for structure-consistent, safety-aware fine-tuning.

**Strengths:**

The paper reframes “post-tuning safety drop” as a dataset-side phenomenon and isolates three concrete levers, such as answer structure, identity calibration, and role-play, rather than attributing degradation to overfitting or model drift.

The study runs controlled interventions (reformatting answers; toggling identity statements; adding or removing role-play) across multiple datasets and base models, then tracks safety alignment changes with clear before/after comparisons, supporting causal claims rather than mere correlations.

Key concepts are defined with care (e.g., “identity calibration” vs “role-play”), the hypotheses are motivated from observed dataset factors, and results are summarised into concrete guidance that ties findings to practitioner decisions in data construction and RM selection.

The work has an immediate impact for anyone fine-tuning aligned LLMs: small formatting and identity cues in training data can shift safety outcomes, and widely used RMs may not predict those shifts.

**Weaknesses:**

The paper could add a small human evaluation (e.g., 100–200 samples) on safety preservation vs. degradation to verify whether automated jailbreak metrics align with human judgment.

Reformatting effects may confound variables: increased verbosity, hedging, or normative phrasing (e.g., “cannot assist”) might themselves cause improved safety, rather than the formatting alone.


Identity calibration may be entangled with lexical or stylistic cues beyond the explicit “As an AI…” statement, so isolating its effect more carefully would strengthen the conclusions.

The claim that “role-play harms safety” is under-specified, such as persona type (e.g., doctor, hacker, safety officer), stance (pro-social vs. adversarial), and instruction strength, which likely interact and should be analysed separately.

The evaluation scope is limited to a few open-source models and datasets; it remains unclear if the findings generalise to frontier closed models, multilingual data, or safety-critical domains such as medical or legal contexts.

The paper shows that common reward models fail to select safer data, but the root cause is unclear. It could be from domain shift, underrepresentation of safety signals in RM training, or sensitivity to instruction style.

**Questions:**

See the weakness, please.

---

### Official Review · Reviewer_SxDe · 2025-11-01

**Soundness:** 3
**Presentation:** 3
**Contribution:** 2
**Rating:** 4
**Confidence:** 4

**Summary:**

The paper empirically studies why safety-aligned LLMs can lose safety when they are later instruction-tuned on **purely benign** downstream datasets. It shows that aligned models are “picky”: small changes in **answer structure** (e.g., Markdown, itemized format), **identity calibration** (responses that explicitly say “as an AI/language model”), and **role-play instructions** can either preserve or significantly **degrade** safety against jailbreak attacks after fine-tuning. The authors further evaluate several state-of-the-art open-source reward models and find that, although they can separate obviously good/bad items within a dataset, they are **unreliable for pairwise selection** and often give *lower* scores to reformatted data that actually makes the LLM safer, revealing a mismatch between RM scoring and true safety preferences. The paper concludes with practical guidance on building safety-aware instruction-tuning datasets and on choosing reward models that better match the target LLM’s post-training signals.

**Strengths:**

**S1: Dual-perspective analysis of alignment drift.**
The paper approaches the problem from two complementary angles—(i) what happens inside the alignment / instruction-tuning pipeline itself, and (ii) how reliable current reward models are in preserving that alignment—making the overall argument quite convincing. This dual view gives the community a clearer picture of where alignment can break (data, formatting, RM) and why seemingly benign fine-tuning can still hurt safety.

**S2: Insightful findings on identity calibration and role-play.**
By separating “identity calibration” style responses (which tend to preserve safety) from “role-play / persona” style prompting (which can erode it), the paper offers a concrete and actionable observation for dataset curators and practitioners. This is a useful nuance that is often glossed over in existing safety-tuning work.

**S3: Coherent use of reformatting throughout the paper.**
The reformatting methodology introduced earlier is actually reused in $\S$5.2, which makes the empirical story self-consistent and avoids the feeling of disconnected experiments.

**S4: Practical guidance for the community.**
Section 6 (“Guidance for Safety-Aware Fine-Tuning”) is genuinely helpful: it translates the empirical observations into concrete data-construction and fine-tuning recommendations, which makes the paper more than a diagnostic study and increases its impact for practitioners.

**Weaknesses:**

**W1: Limited novelty.**
The main contribution is largely analytical, and the “automatic reformatting pipeline” is essentially a deterministic pre-processing script; compared with existing work on alignment-data curation and safety-preserving instruction tuning, the incremental innovation is modest.

**W2: No assessment of utility–safety tradeoffs.**
The paper reports how reformatting, role-play, and identity calibration affect safety metrics, but it does not evaluate whether these paradigms hurt the model’s primary task performance. As a result, it is unclear whether the observed safety gains come at the cost of degraded utility.

**W3: Lack of realistic downstream validation.**
All experiments follow the “benign fine-tuning → jailbreak evaluation” pattern; there is no end-to-end, high-stakes scenario (medical, code, enterprise assistant) demonstrating that such formatting shifts would actually make a deployed system unsafe, so the impact remains somewhat hypothetical.

**W4: Inconsistent effect of role-play.**
As noted around Lines 343–345, the influence of role-play on alignment is model-dependent and not stable across architectures, which makes it hard for practitioners to know whether they should adopt or avoid this paradigm; the paper’s guidance on this point is therefore weaker.

**Questions:**

Clarification on notation (Lines 205–206): The paper introduces $SA_g$ and $SA_a$ at this point, but it is not clear what these symbols stand for, how they are computed, and how they differ from each other. Please define both terms explicitly.

---

### Meta-Review · Area_Chair_4kDj · 2026-01-03

**Summary:**

This paper provides an analytical study on the key factors influencing safety alignment.

- The reviewers raised a number of concerns, including
- Limited novelty.
- No assessment of utility–safety tradeoffs.
- Lack of realistic downstream validation.
- Inconsistent effect of role-play.
- Lack of human evaluation.
- Reformatting may lead to confounding factors.
- The evaluation scope is limited to a few open-source models and English-only datasets.
- Only Lora is considered.

**Reviewer Concerns:**

No rebuttal was provided.

**Reviewer Scores:**

No rebuttal was provided.

---

### Decision · Program_Chairs · 2026-01-26

Reject